# Integrated proteomics reveals autophagy landscape and an autophagy receptor controlling PKA-RI complex homeostasis in neurons

Xiaoting Zhou[1,2,3], You-Kyung Lee[1,2], Xianting Li[1,2], Henry Kim[1,2], Carlos Sanchez-Priego[2,4], Xian Han[5,6,7], Haiyan Tan[5,6,8], Suiping Zhou[8], Yingxue Fu[5,6,8], Kerry Purtell[1,2], Qian Wang[1,2], Gay R. Holstein[1,2], Beisha Tang[9,10], Junmin Peng[5,6,8] ✉, Nan Yang[2,4] ✉ & Zhenyu Yue[1,2,11] ✉

Autophagy is a conserved, catabolic process essential for maintaining cellular homeostasis. Malfunctional autophagy contributes to neurodevelopmental and neurodegenerative diseases. However, the exact role and targets of autophagy in human neurons remain elusive. Here we report a systematic investigation of neuronal autophagy targets through integrated proteomics. Deep proteomic profiling of multiple autophagy-deficient lines of human induced neurons, mouse brains, and brain LC3-interactome reveals roles of neuronal autophagy in targeting proteins of multiple cellular organelles/pathways, including endoplasmic reticulum (ER), mitochondria, endosome, Golgi apparatus, synaptic vesicle (SV) for degradation. By combining phosphoproteomics and functional analysis in human and mouse neurons, we uncovered a function of neuronal autophagy in controlling cAMP-PKA and c-FOS-mediated neuronal activity through selective degradation of the protein kinase A - cAMP-binding regulatory (R)-subunit I (PKA-RI) complex. Lack of *AKAP11* causes accumulation of the PKA-RI complex in the soma and neurites, demonstrating a constant clearance of PKA-RI complex through AKAP11-mediated degradation in neurons. Our study thus reveals the landscape of autophagy degradation in human neurons and identifies a physiological function of autophagy in controlling homeostasis of PKA-RI complex and specific PKA activity in neurons.

Macroautophagy (herein referred to as autophagy) is a lysosome-dependent degradation pathway. Mammalian cells require autophagy to maintain cellular homeostasis and respond to various stresses and injuries through digestion and recycling. During autophagy, cytoplasmic components are engulfed in a double-membraned vesicle known as the autophagosome, which is shuttled to the lysosome for degradation[1]. While autophagy degrades intracellular components in the non-selective manner in response to nutrient starvation, autophagy can also selectively degrade cytoplasmic proteins and organelles under basal conditions. More specifically, Atg8-family proteins such as LC3A/B, which are conjugated on the growing autophagosome membrane, selectively recognize autophagy cargos[2]. The identification of specific interactions between Atg8-family proteins and autophagy cargos has enabled the utilization of

affinity purification and proximity labeling to profile selective autophagy substrates in cancer cell lines[3–5].

In contrast to most mammalian cell types, neurons are post-mitotic, long-living cells of the central nervous system (CNS). To maintain longevity and physiological function, neuronal autophagy removes and recycles misfolded proteins, protein aggregates, and injured organelles that would otherwise be detrimental to the cell. In fact, when essential autophagy-related genes (ATGs) *Atg5* and *Atg7* are conditionally deleted in mouse brains, cerebral and cerebellar neurons degenerate, causing motor dysfunctions in mice[6,7]. Using mice with disrupted *Atg7* expression in specific neuron types, we previously demonstrated a compartmental function of autophagy in neurons by maintaining axonal homeostasis and preventing axonopathy[8,9]. Indeed, autophagy biogenesis can occur in the axon and participates in axonal transport to remove dysfunctional organelles and protein aggregates[10–12]. Furthermore, autophagy was shown to regulate synaptic activity[13–15]. The significance of autophagy in the CNS is further underscored in humans, where a recent study has identified biallelic, recessive variants in human *ATG7* that results in impaired autophagic flux and complex neurodevelopmental disorders[16]. Furthermore, dysfunctional autophagy may contribute to major neurodegenerative diseases[17–19]. However, the physiology of autophagy in human neurons and detailed mechanisms for how its disruption leads to various neurological disorders remains poorly understood.

To understand the autophagy targets, multiple studies performed systematic analysis to dissect autophagy cargo in non-neuronal cells[4,20–26]. However, the investigation of neuronal autophagy cargo, particularly in humans, remains limited. Profiling the differentially accumulated proteins from *Atg5*-deficient primary mouse neurons has revealed the function of autophagy in regulating axonal endoplasmic reticulum (ER) and luminal $Ca^{2+}$ stores[27]. Examination of isolated autophagic vesicles (AVs) from mouse brains using differential centrifugation showed the presence of mitochondrial fragments and synaptic proteins in neuronal autophagosomes[10].

Herein we present a systematic investigation of human neuronal autophagy cargo by using human pluripotent stem cell (PSC)-induced neurons (iNeurons) through integrated proteomics and functional analysis. To enrich autophagy cargos, we suspended autophagy in iNeurons using CRISPR-interference technology to knockdown *ATG7* or *ATG14*, both of which are essential for autophagy functions. By combining the study of the neurons from multiple lines of mouse autophagy deficient brains, our data reveals the landscape of autophagy degradation in neurons and highlights selective sets of proteins from multiple organelles as potential cargos of neuronal autophagy. Moreover, we identified a function of autophagy in controlling PKA-RI complex homeostasis through autophagy receptor AKAP11 and regulating neuronal activity. Our study thus provides a global view of autophagy degradation in human neurons and an insight into the mechanisms of neurological disorders linked to autophagy deficiency.

## Results
### Proteomic profiling of enriched cellular pathways in autophagy-deficient human induced neurons
Previous studies demonstrated that autophagy is constitutively active in neurons; basal autophagy constantly degrades cellular cargo to maintain neuronal homeostasis and protect neurons[8,28]. To systematically investigate neuronal autophagy targets under basal conditions, we suspended autophagic flux in neurons, performed integrated proteomic profiling of accumulated proteins, and identified autophagy cargo, receptors, and targeted pathways (Fig. 1a). To ensure the rigor of the analysis and isolate changes specific to autophagy inhibition, we elected to disrupt two essential autophagy genes, *ATG7* and *ATG14*, separately in human iNeurons as well as *Atg7* and *Atg14* in mouse brains in a neuron-specific manner. ATG7 is an E1-like enzyme, which is required for the lipidation of LC3 and Atg5-Atg12-Atg16 conjugation[29].

ATG14 is an essential subunit of the Beclin1-PI3K complex, which facilitates the tethering and formation of phagophore[30,31]. Blocking either *ATG7* or *ATG14* leads to disruption of autophagosome formation and the consequent autophagy cargo accumulation in the cells. We posit that examination of the results from the inactivation of the two functionally distinct autophagy genes facilitates the identification of the proteins related to autophagy degradation. We also performed transcriptomic analysis by RNA sequencing (RNA-seq) to discern the change of protein levels not caused by transcriptional alteration. Furthermore, we investigated the interactome of autophagy protein LC3 in the brains by including the green fluorescence protein-tagged LC3 (GFP-LC3) transgenic mice[32] in our study (Fig. 1a).

Autophagy in human neurons is poorly characterized. We sought to investigate autophagy in human glutamatergic iNeurons, representing the most common neuron type in the CNS. We used our previously established approach to generate iNeurons by transiently expressing the transcription factor neurogenin 2 (Ngn2)[33]. Fusion of a catalytically inactive dCas9 to the KRAB repressor domain enables robust knockdown (KD) of endogenous genes in iNeurons[34]. To repress *ATG7* or *ATG14* expression, we transduced human PSCs with a lentiviral construct expressing dCas9-KRAB and sgRNA targeting *ATG7* or *ATG14*. Reduction of ATG7 or ATG14 protein levels was robust in PSCs and iNeurons (Fig. 1b, Supplementary Fig. 1a, b). At 6 weeks post-differentiation (PD), the iNeurons had obtained stable expression of dendrite and synapses markers (Fig. 1c), and robust electrophysiological properties and formed functional synapses[33].

We next profiled the proteome of *ATG7* and *ATG14* KD iNeurons (6-week PD) by the multiplexed tandem mass tag (TMT)-based, two-dimensional liquid chromatography (LC/LC) and tandem mass spectrometry (MS/MS) analysis[35,36]. Isobaric labeling methods like TMT have become an important tool for in-depth proteomics profiling and are increasingly used for the multiplexed analysis of samples, offering high throughput and consistency[37,38]. A known limitation of isobaric tags is the co-isolation of target peptide ions with co-eluting peptides during MS2-based quantification, which can introduce considerable noise and potentially skew quantitative ratios. Fortunately, this ratio suppression also reduces experimental variations, minimally affecting the statistical analysis of significant proteome alterations after z-transformation[39,40]. The ratio suppression can be alleviated by approaches such as intensive peptide fractionation, employing a narrower MS2 isolation window, and computational correction[37]. Moreover, the MS3 approach can virtually eliminate these inaccuracies, but it necessitates additional duty cycles and specific MS configurations, along with low-resolution MS2 data for identification, which negatively impact peptide/protein identification rates[38,41]. To balance the pros and cons of the TMT method, we have optimized the method by employing extensive high-resolution LC/LC-MS/MS, which significantly improves proteome coverage and reduces ion suppression effects during quantification[42–46]. We identified approximate 8000 proteins across all the human iNeuron samples (Supplementary Data 1b), demonstrating the high coverage of the proteome. Principal-component analysis (PCA) revealed reproducible replicate data, with the variance driven mainly by the difference in ATG gene expression. Similarly, clustering samples by protein expression similarity showed two main groups representing the mutant and control iNeurons (Supplementary Fig. 1c, d). We identified 1460 and 1172 differentially expressed proteins (DEPs) in *ATG7* KD and *ATG14* KD iNeurons, respectively ($p$ value < 0.05, |$\log_2$FC|>2- fold standard deviation (SD), SD = 0.18, Supplementary Data 1b). Our quantitative analyses reveal 1113 and 637 upregulated DEPs of proteins in *ATG7* KD and *ATG14* KD iNeurons, respectively, as shown in volcano plot analysis (Supplementary Data 1b, Fig. 1d, e). Gene ontology (GO) enrichment analysis of the upregulated DEPs highlighted cellular pathways associated with endoplasmic reticulum (ER), Golgi apparatus, synaptic vesicle (SV), mitochondria, endosome, and lysosome in both *ATG7* KD and *ATG14*

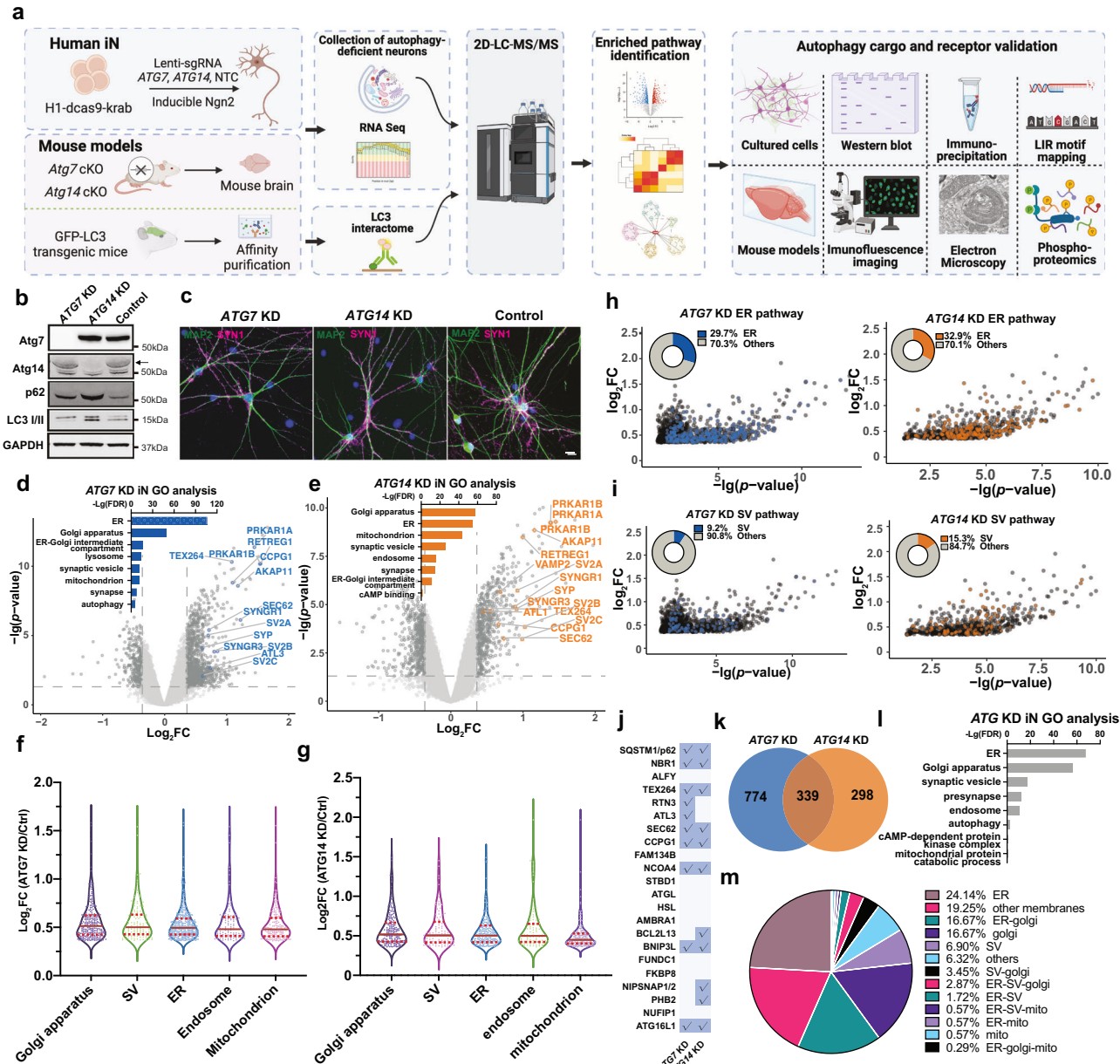

**Fig. 1 | Proteomics analysis of human induced neurons (iNeurons) with *ATG7* or *ATG14* knock-down (KD). a** An overview of the workflow for the integrated study of autophagy cargo in human iNeurons and mouse brains. Created with BioRender.com. **b** H1-dCas9-KRAB stem cells were used for the knock-down (KD) of *ATG7* or *ATG14* (arrow) and the block of autophagy. **c** Immunofluorescence images of *ATG7* KD and *ATG14* KD human iNeurons (6-week-old) co-stained with anti-MAP2 (green) and synapsin1 (SYN1, magenta) antibodies. Scale bar, 20 μm. **d-e** Volcano plot of the differentially expressed proteins (DEPs) detected by LC/LC-MS/MS in *ATG7* KD (**d**) and *ATG14* KD (**e**) human iNeurons compared with control. Bar graph shows significant Gene Ontology (GO) annotations ($p$ value < 0.05, Log$_2$FC > 2 SD, SD = 0.18). FDR was calculated with the moderated two-sided *t*-test in the LIMMA package (R Studio). GO analyses were performed in g:Profiler (https://biit.cs.ut.ee/gprofiler/gost). **f, g** Violin plots of DEPs from the GO analysis in *ATG7* KD and *ATG14*

KD iNeurons, respectively. Each dot represents one protein. Solid red bars indicate the median Log$_2$FC, and the dashed bars specify the 25th and 75th interquartile range. Plot of Log$_2$FC and -Lg (FDR) for ER (**h**) or SV (**i**) proteins of *ATG7* KD (blue, left) and *ATG14* KD (orange, right) ($p$ value < 0.05, Log$_2$FC > 2 SD, SD = 0.18) human iNeurons. Pie chart shows the percentage of ER or SV DEPs and the rest from the mutant human iNeurons. **j** A table of known autophagy receptors and those detected (check marks) in proteomic analysis ($p$ value < 0.05, Log$_2$FC > 2 SD, SD = 0.18). The list of autophagy receptors was manually collated through literature searches. Venn diagram (**k**) shows the overlap between the 339 DEPs of *ATG7* KD and *ATG14* KD iNeurons ($p$ value < 0.05, Log$_2$FC > 2 SD, SD = 0.18) and GO enrichment analysis of shared DEPs (**l**). **m** Pie chart shows the percentage of proteins categorized into different organelles or pathways based on the dataset from **l**.

KD iNeurons (Fig. 1d, e). By plotting the fold increase of the upregulated DEPs, we obtained the mean values and inferred the abundance of the proteins for different organelles or pathways (Fig. 1f, g). Golgi, SV and ER DEPs noticeably displayed the greatest abundance in both *ATG7* and *ATG14* KD iNeurons relative to DEPs of mitochondria and endosome. Indeed, nearly one-third of the total upregulated DEPs are ER-related proteins in both *ATG7* and *ATG14* KD iNeurons (Fig. 1h),

while 9.2% and 15.3% of upregulated DEPs are SV proteins in *ATG7* KD and *ATG14* KD iNeurons, respectively (Fig. 1i). Among the upregulated DEPs in *ATG7* KD and *ATG14* KD iNeurons, at least thirteen were known autophagy receptors as previously reported[47–49], including five known ER-phagy receptors (Fig. 1j). The identification of the previously described autophagy receptors validated the feasibility and robustness of our approach.

To ask whether the increased levels of the proteins are due to transcriptional upregulation of their encoding genes, we performed RNA-seq analysis of *ATG7* KD iNeurons. Strikingly, *ATG7* was found as the only differentially expressed gene (DEG) (FDR < 0.05) (Supplementary Fig. 1h). Furthermore, we found that 339 upregulated DEPs are shared between *ATG7* KD and *ATG14* KD iNeurons (Fig. 1k), suggesting that they are unlikely *ATG7* or *ATG14* - specific but related to autophagy degradation. Consistently, these shared proteins were enriched in ER, Golgi apparatus, mitochondria, and SV-related pathways (Fig. 1l, m). Noticeably ER proteins are top ranked as shown in both GO analysis (FDR) and the percentage of ER proteins in the upregulated DEPs (Fig. 1l, m). Interestingly, we found the enrichment of "cAMP-dependent protein kinase complex" among the shared DEPs (Fig. 1l).

Furthermore, we identified 347 and 535 down-regulated proteins in *ATG7* KD and *ATG14* KD iNeurons, respectively ($p < 0.05$, $Log_2FC < -2SD$). 106 down-regulated DEPs are shared by the two mutant cells (Supplementary Fig. 1f). We found significant enrichment of the down-regulated pathways related to protein binding, synapse, and neuron development (Supplementary Fig. 1g).

## Impaired clearance of selective ER and SV proteins in autophagy-deficient human iNeurons

Our quantitative proteomics suggest that many ER proteins are potential autophagy cargo in human neurons, as indicated by high-ranking GO terms and abundance analysis in human iNeurons (Fig. 1d, e and Supplementary Data 1b). Multiple ER membrane proteins, including RTN3, ATL1, SEC62, and TEX264, function as receptors responsible for the selective degradation of the ER by autophagy (ER-phagy)[4,50–52]. Immunoblot analysis confirmed that these ER-phagy receptors, along with ER resident proteins REEP5 and Calnexin (CNX), are increased or have a trend of increase in *ATG7* KD and *ATG14* KD iNeurons (Fig. 2a, c and supplementary Fig. 1e). A previous proteomic study classified ER proteins into four groups, such as ER membrane, ER lumen, ER-Golgi intermediate compartment (Ergic/cisGolgi), and ER curvature[53]. An overall increase of all categories was identified in both mutant iNeurons (Fig. 2e, f). Immunofluorescence (IF) analysis showed a remarkable accumulation of CNX-labeled inclusions (arrows) particularly in the neurites of *ATG7* KD iNeurons (Fig. 2g, h), suggesting an expansion of ER network in the processes. Using an ER-phagy reporter, which expresses an N-terminal ER signal sequence followed by tandem monomeric RFP and GFP sequences and the ER retention sequence KDEL[4], we found that the signal of the RFP was significantly reduced in *ATG7* KD iNeurons upon nutrient-withdrawal, implying an impairment of ER-phagy (Supplementary Fig. 1i–k).

We next examined the accumulation of cargo candidates associated with SV. Significant increase of synaptogyrin1 (SYNGR1), synaptogyrin3 (SYNGR3), synaptophysin1 (SYP1), SV2A, and SV2C in autophagy-deficient iNeurons was confirmed by immunoblotting (Fig. 2b, d and Supplementary Figs. 1e and 2a). Further inspection of cargo candidates through SynGO analysis showed greater number of upregulated DEPs associated with presynapse than with postsynapse (Fig. 2i, j), consistent with increased abundancy of autophagy cargo in autophagy-deficient presynaptic terminals[10,15,32]. IF staining revealed a significant increase in the size and density of SV2A-, SYP1-, and SYNGR3- associated puncta in *ATG7* KD iNeurons (Fig. 2k, l), corroborating the clearance blockage of the above presynaptic proteins. Treatment of iNeurons with bafilomycin A1 caused an increase or trend of increase of SYNGR3, SV2A, SV2C, SYP1, and AKAP11 protein levels, supporting their degradation through autophagy-lysosome (Supplementary Fig. 2b, c).

Collectively, our findings indicate that specific sets of ER and SV proteins require autophagy to maintain their homeostasis or degradation in human neurons under basal conditions. ER-phagy is impaired in human *ATG7* KD iNeurons.

## Proteomic profiling of enriched cellular pathways in autophagy-deficient neurons from mutant mouse brains

To validate our findings from human iNeurons, we sought to profile the targets of neuronal autophagy in vivo by using mouse brains. We selectively deleted *Atg7* or *Atg14* in CNS neurons using the combination of synapsin1 promoter-driven Cre recombinase mice (SynCre) and the *Atg7*[f/f] or *Atg14*[f/f] mice to generate conditional knock-out (cKO) mice. Immunoblot analysis of whole brain lysates from *Atg7*[f/f]-SynCre (*Atg7* cKO) and *Atg14*[f/f]-SynCre (*Atg14* cKO) mice confirmed the reduction of ATG7 and ATG14 protein levels, respectively, and an increase of autophagy substrate p62 and ubiquitinated protein levels (Supplementary Fig. 3a–d). Both autophagy-deficient mice exhibited a progressive reduction in body weight (Fig. 3a, b) and viability (Fig. 3c, d) over time, while *Atg14* cKO mice are more vulnerable than *Atg7* cKO mice. *Atg7* cKO and *Atg14* cKO brains (6–8-week-old) were then subjected to the TMT-LC/LC-MS/MS analysis. We profiled 9638 and 6252 proteins from *Atg7* cKO and *Atg14* cKO mice, respectively (Supplementary Data 1c and Supplementary Data 1d). Most of the DEPs showed an increase in abundance (as opposed to a decrease) in the mutant brains (Fig. 3e, f, Supplementary Fig. 3e, f), revealing a profound impact of autophagy deficiency on protein degradation. Indeed, 459 and 343 proteins were increased in *Atg7* cKO and *Atg14* cKO mouse brains compared to wild-type, respectively ($p < 0.05$, | $Log_2FC$ | >2-SD, SD = 0.1 and 0.16 for *Atg7* cKO and *Atg14* cKO, respectively) (Fig. 3e, f, Supplementary Data 1c and Supplementary Data 1d). Like the findings in human iNeurons (Fig. 1), the fraction of proteins whose abundance increased in the mutant mouse brains are enriched for GO terms, such as ER, mitochondria, Golgi apparatus, endosome, SV, autophagosomes, and cAMP-dependent kinase (PKA) (Fig. 3e, f, Supplementary Fig. 3g, h). Indeed, the upregulated DEPs of ER proteins account for 37% and 31% of the total upregulated DEPs from *Atg7* cKO and *Atg14* cKO brains, respectively (Fig. 3i). We found that 102 upregulated DEPs are shared between *Atg7* cKO and *Atg14* cKO brains (Fig. 3g), while ER components are the most significant in GO term analysis (Fig. 3h). We detected increased abundance of SV proteins as well as cPKA pathways in both *Atg7* cKO and *Atg14* cKO mouse brains (Fig. 3e, f, i, j). Known ER-phagy receptors, such as TEX264, SEC62, and FAM134B, were found among upregulated DEPs in *Atg7* cKO or *Atg14* cKO mouse brains (Fig. 3k). Furthermore, transcriptomic analysis using RNA-seq of *Atg7* cKO brains did not reveal an overlap between the upregulated DEGs and DEPs in *Atg7* cKO brains (Supplementary Fig. 3e, i). For example, upregulated DEGs are activated glial marker genes, such as Gfap, C1qc, Clec7a, and Tyrobp (Supplementary Fig. 3i). Thus, the increase of many proteins identified in *Atg7* cKO brains is unlikely due to the increase in transcriptional upregulation.

Consistent with human iNeuron results (Figs. 1 and 2), immunoblot analysis demonstrated an increase or a strong tendency of increase in the abundance of ER proteins and ER-phagy receptors, such as RTN3, ATL1, SEC62, CNX and REEP5, in *Atg7* cKO and *Atg14* cKO brains; and similar observations were found for SV proteins, including SV2A, SV2C, SYNGR1, SYNGR3, and SYP1 (Fig. 3l, m, Supplementary Fig. 3j, k) but to a lesser degree. To determine if the ER protein accumulation is associated with ER structure alteration in mutant neurons, we performed electron microscopy (EM) analysis of *Atg7* cKO mouse brains and found a remarkable expansion or stacking of tubular ER in the axons, corroborating with the increased levels of a specific set of ER proteins, such as tubular ER proteins RTN3 and REEP5, in mutant neurons (Fig. 3n).

Taken together, the results from mouse brains validated the observations in human iNeurons that basal autophagy maintains the homeostasis of proteins of multiple pathways and/or organelles in neurons, including ER, Golgi, mitochondria, SV, endosomes, lysosomes, and cAMP-PKA kinase complex.

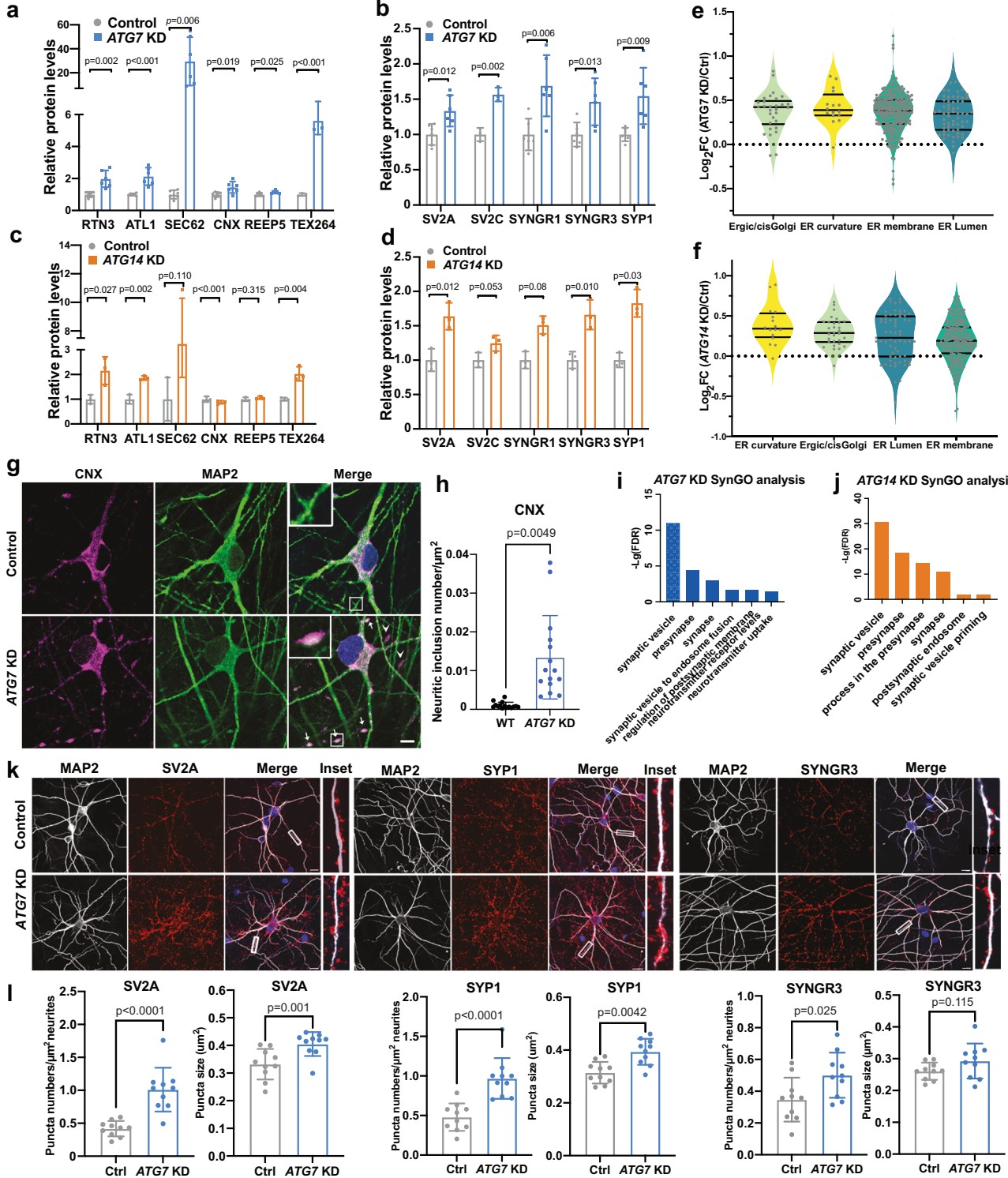

## Investigation of LC3-interactome in autophagy-deficient neurons

LC3 is the best-characterized mammalian Atg8 homolog and it contains the LC3-interation region (LIR) that mediates cargo recruitment for degradation[54]. The binding of LC3 to the cargo does not necessarily depend on the lipidation of LC3[47]. GFP-LC3 transgenic mice were used to investigate LC3-binding proteins in mouse brains[32]. Under basal conditions, however, GFP-LC3 is rarely detected in puncta or association with autophagosomes in the CNS neurons[32,55], possibly due to rapid cleavage of GFP-LC3 or turnover of autophagosomes[56]. To facilitate the identification of LC3-interacting cargos using anti-GFP

affinity isolation in neurons, we inhibited neuronal autophagy in GFP-LC3 transgenic mice by crossing GFP-LC3 mice with the *Atg7*[f/f]-Syn-Cre mice (GFP-LC3; *Atg7* cKO) and then performed GFP-pull-down from GFP-LC3; *Atg7* cKO brains (Fig. 4a). Immunoblot and silver staining confirmed the enrichment of proteins with GFP affinity purification, particularly in the GFP-LC3; *Atg7* cKO mice (Supplementary Fig. 4a). Mass spectrometry analysis was performed to identify GFP-LC3 binding proteins remaining on the beads. The hierarchical clustering analysis confirmed similarity across biological replicates (Supplementary Fig. 4b). Next, we filtered out non-specific binding proteins by comparing GFP-LC3 and non-transgenic mice and focused on interactions

**Fig. 2 | Validation of the DEPs of ER and SV related proteins in autophagy-deficient human iNeurons.** Quantification of ER (**a**, **c**) and SV (**b**, **d**) proteins in *ATG7* KD and *ATG14* KD iNeurons, respectively. Relative protein levels were normalized to the loading control β-actin. *N* = 6 (for RTN3, ATL1, SEC62, CNX, REEP5, SV2A, SYNGR1, SYNGR3, and SYP1 for *ATG7* KD iNeurons), and *n* = 3 (for TEX264 and SV2C) biologically independent replicates for the proteins. The controls' means were set to 1. Two-sided unpaired t-test. Ns, no significance. **e**, **f** Violin plots of ER sub-compartment proteins enriched in ER curvature, ER-Golgi intermediate compartment (Ergic/cisGolgi), ER lumen, ER membrane in *ATG7* KD iNeurons. Each dot represents one protein. Solid black bars indicate the median Log$_2$FC, and the dashed bars specify the 25th and 75th interquartile range. **g** Immunofluorescence images of Calnexin (CNX, magenta) and MAP2 (green) staining in *ATG7* KD and control iNeurons. Insets, magnified images of the boxed region. **h** Quantification of CNX+ large inclusions per μm$^2$ of total neuronal dendrite in *ATG7* KD iN and control iN. Two-sided unpaired t-test; Scale bar, 10 μm. (*n* = 15). **i** Bar graph of top ranked GO terms generated by using SynGO from *ATG7* KD vs. control iN (*p* value < 0.05, Log$_2$FC > 2 SD, SD = 0.18). **j** Bar graph of top ranked GO terms generated by using SynGO from *ATG14* KD vs. control iN. **k** Immunofluorescence images of *ATG7* KD human iNeurons stained with anti-SV2A, anti-synaptophysin1 (SYP1), and anti-synaptogyrin3 (SYNGR3), and anti-MAP2 antibodies. The highlighted areas were enlarged in inset. Scale bar, 10 μm. (3 biological replicates, *n* = 10 per batch). **l** Quantification of synaptic "puncta" density and size labeled with SV2A, SYP1, and SYNGR3 from **k**. Unpaired t-test; Scale bar, 10 μm. (3 biological replicates, *n* = 10 per batch). All data are shown as mean ± SEM.

that were altered in *Atg7* cKO. Compared with *Atg7* WT mice, *Atg7* cKO mice had differential GFP-LC3 interaction in 600 proteins (*p* < 0.05, Log$_2$FC > 2 SD (SD = 0.21)), while 73.7% were increased and 26.3% were decreased (Fig. 4b, Supplementary Data 1e).

We then performed pairwise comparisons between proteins with increased GFP-LC3 interaction in *Atg7* cKO brains and upregulated DEPs in *ATG7* KD iNeurons, *ATG14* KD iNeurons, *Atg7* cKO mouse brain or *Atg14* cKO mouse brains. The results revealed a considerable overlap of the increased protein levels (Fig. 4c, d, e, f). GO enrichment analyses revealed that the molecular functions and cellular component of the overlapping proteins primarily comprised ER, Golgi trafficking, ER stress response, SV cycle, response to unfolded proteins, the autophagosome, and cAMP-dependent kinase (PKA) (Fig. 4c, d, e, f). The reproducible enrichment of the PKA pathways as shown in the above multiple datasets is particularly intriguing, as we previously reported that autophagy selectively degrades PKA-RI subunits and regulates PKA activity through the AKAP11 receptor in tumor cells[57]. Unsurprisingly, autophagy receptors (p62/SQSTM1, NBR1), ER-phagy receptors (TEX264, RTN3, SEC62, CCPG1), and mitophagy receptors (FUNDC1, NIPSNAP1/2) were detected in GFP-LC3 interactome validating our detection system (Supplementary Fig. 4d). By integrating GO enrichment pathways results from the above five datasets, we observed that the ER pathway is the most significant across most of the datasets, supporting the notion that ER is a prominent target of autophagy in neurons (Fig. 4g). The ER membrane is the most enriched among all categories of ER related proteins, which are shared by the five datasets (Fig. 4h, j). Furthermore, we identified 45 proteins, which are shared by the four autophagy-deficient neuron datasets, and 15 shared by all five proteomic datasets (with LC3-interactome) (Fig. 4i). These shared proteins include known autophagy cargo, such as SQSTM1, SEC62, NBR1, TEX264, PRKAR1A, and PRKAR1B. By performing protein-protein interaction (PPI) network analysis with selective upregulated DEPs shared by *ATG7* KD and *ATG14* KD iNeurons (Fig. 1k), we found specific and extensive interconnectivity among functional modules representing autophagy, ER, Golgi apparatus, SV, PKA pathway and mitochondrion (Fig. 4k, Supplementary Fig. 4e). Many components of the above modules were validated for their dependence on autophagy for clearance based on this study, including AKAP11, PRKAR1A, REEP5, ATL1, RTN3, SV2, SYNGR3, SYP (highlighted in red) (Fig. 4k).

We next validated the interaction of LC3 with ER, SV, and PKA pathway proteins using the GFP-LC3 transgenic brains and iNeurons stably expressing GFP-LC3 (Supplementary Fig. 4c). Co-IP with GFP antibody confirmed interactions of LC3 with SV proteins (SV2A and SYNGR3) and AKAP11 in the mouse brain (Fig. 4l). Moreover, we treated GFP-LC3 expressing iNeurons with chloroquine to block autophagy flux and performed IF analyses. Chloroquine apparently enhanced the number of GFP-LC3 puncta (indicating autophagosomes), which partially co-localize with the endogenous SV2A, and SYNGR3 (Fig. 4m). These observations suggest that many LC3-binding proteins identified above are recruited to autophagosomes via LC3 for autophagy degradation in iNeurons.

## Autophagy degrades AKAP11 and RI proteins and regulates PKA and neuronal activity in human iNeurons

The above proteomics profiling and LC3-interactome analysis suggest that autophagy may selectively degrade the components of the cAMP-PKA pathway in human and mouse neurons. The PKA holoenzyme is a tetramer consisting of two regulatory subunits (R-subunits, RI and RII) and two catalytic subunits (PKAc). Each R subunit binds and inhibits a catalytic subunit at the resting state[58]. cAMP binding to the R-subunits releases and activates PKAc[59,60].

Given the important functions of cAMP-PKA in neurons[61,62], we next explored the role of autophagy in the regulation of PKA-dependent neuronal activity. Immunoblotting revealed that the protein levels of AKAP11, RI, and PKA Catalytic α subunit (PKA-Cα), but not RII, were significantly increased in the two types of autophagy-deficient iNeurons (Fig. 5a). Blockage of autophagic flux by chloroquine leads to the accumulation of RI proteins colocalized with autophagosome markers GFP-LC3 and p62 (Fig. 5b–d). To investigate whether autophagy regulates PKA-dependent neuronal activity, we analyzed forskolin-dependent phosphorylation of CREB (p-CREB) responses and expression of the immediate early response gene c-FOS in control and *ATG7* KD iNeurons. Immunoblot analysis revealed a reduction of forskolin-induced p-CREB in *ATG7* KD iNeurons (Supplementary Fig. 5a). Intriguingly, we found attenuated forskolin-dependent c-FOS induction (the neuronal activity early response gene), suggesting a decrease of neuronal activity in *ATG7* KD iNeurons (Fig. 5e and Supplementary Fig. 5b). IF analyses showed the reduced staining intensity of p-CREB and c-FOS in the nuclei of *ATG7* KD iNeurons (Fig. 5f, g). Studies of the primary cortical mouse neurons derived from *Atg7* cKO mice showed the similar results as in human iNeurons (Supplementary Fig. 5c).

PKA phosphorylates diverse substrates to regulate critical functions in neurons[63]. We posited that autophagy deficiency could lead to reduced phosphorylation of many PKA substrates (Fig. 5h) due to the aberrant retention of the RI that sequesters the catalytic subunit. We then performed a phospho-proteomics analysis of *ATG7* KD and *ATG14* KD iNeurons (Supplementary Fig. 5d, e). We identified 458 down-regulated phosphorylated protein sites in *ATG7* KD and 217 in *ATG14* KD iNeurons, respectively. There were 86 commonly phosphorylated sites between *ATG7* and *ATG14* KD iNeurons, of which many proteins containing the phosphorylated sites were predicted as PKA substrates (Fig. 5i, Supplementary Data 1f). For example, phosphorylation of PKA substrates, including ADD1, ADD2, and NGFR[64–66], were reduced in *ATG7* KD and *ATG14* KD iNeurons. The proteins with reduced phosphorylation in the autophagy-deficient iNeurons showed strong enrichment in GO terms associated with neuronal functions such as neuronal cellular compartments, neurogenesis, and neuron development (Fig. 5j). By screening PKA consensus motif - containing proteins (R/K-X-X-pS/T)[67] and those with reduced phosphorylation in autophagy-deficient iNeurons, we identified 16 putative PKA substrates in the 86 shared down-regulated phospho-proteins (Fig. 5k). Future studies will be needed to illustrate further their function in mediating neuronal activity.

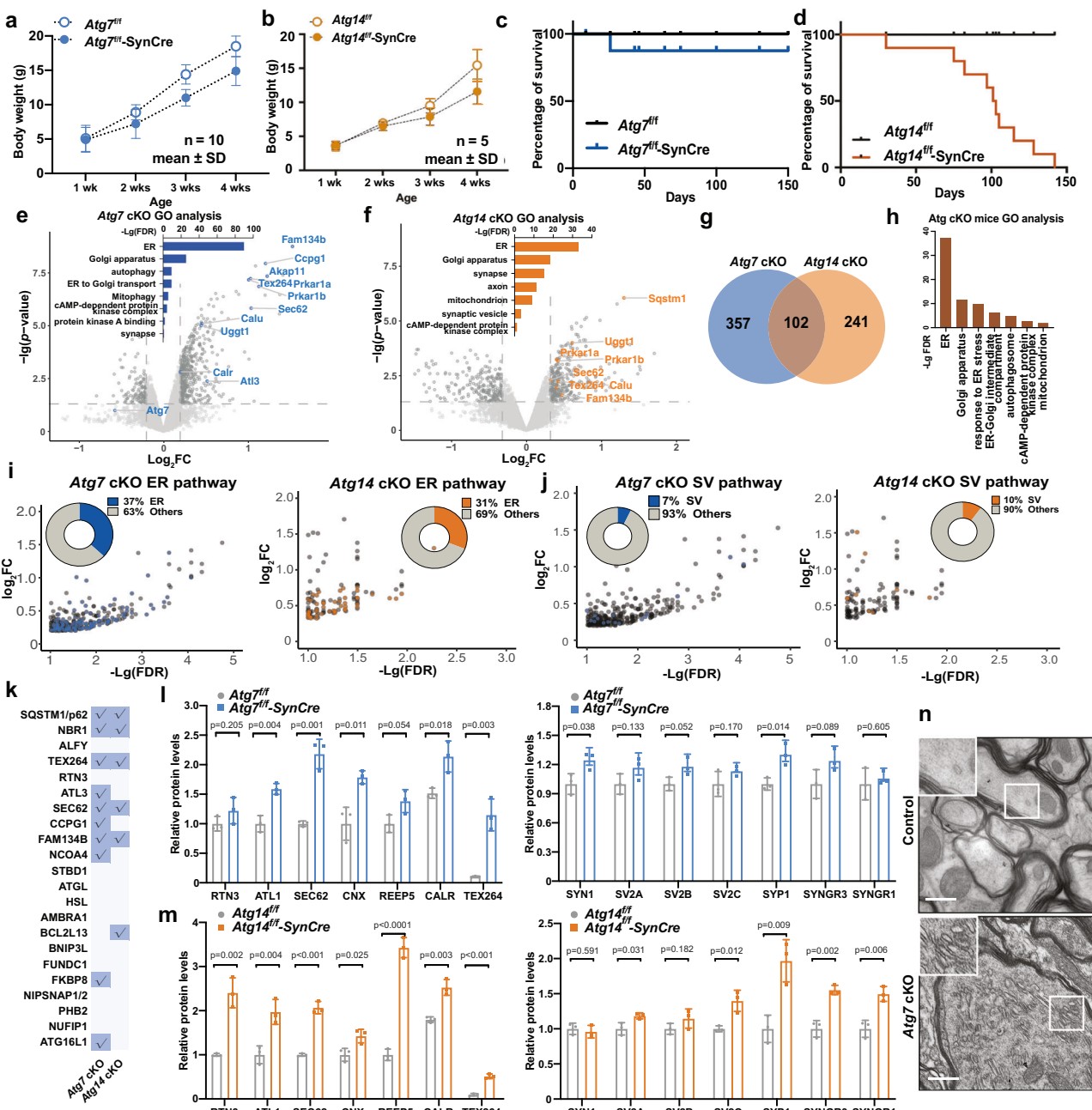

**Fig. 3 | Proteomic analysis of proteins from mouse brains with neuron-specific KO of *Atg7* or *Atg14*.** Graph depicting body weight (grams) of *Atg7*[f/f] and *Atg7*[f/f]-SynCre mice (**a**), *Atg14*[f/f] and *Atg14*[f/f]-SynCre mice (**b**) at different ages. Data points represent mean +/− SD. Kaplan–Meier survival curve of *Atg7*[f/f] and *Atg7*[f/f]-Syn-Cre (**c**), *Atg14*[f/f] and *Atg14*[f/f]-Syn-Cre (**d**) mice at different ages. **e** Volcano plot of the DEPs from proteomic data of *Atg7*[f/f] and *Atg7*[f/f]-SynCre mouse brains. Inset bar graph shows the enriched GO terms associated with DEPs (*p* < 0.05, Log₂FC > 2 SD, SD = 0.1). **f** Volcano plot of the DEPs from proteomic data of *Atg14*[f/f] and *Atg14*[f/f]-Syn-Cre mouse brains. Inset bar graph shows the enriched GO terms among DEPs (*p* < 0.05, Log₂FC > 2 SD, SD = 0.16). **g** Venn diagram shows the overlap between the DEPs of *Atg7* cKO and *Atg14* cKO mouse brains. **h** GO enrichment analysis of 102 DEPs shared between *Atg7* cKO and *Atg14* cKO mouse brains. Plot of Log₂FC and -Lg (FDR) for ER (**i**) and SV (**j**) DEPs from *Atg7* cKO and *Atg14* cKO mouse brains,

respectively. DEPs are indicated by blue (*Atg7* cKO) and orange dots (*Atg14* cKO). Inset circle plots show the percentage of ER DEPs vs. the rest of the DEPs. ER and SV proteins were defined by their GO annotations in gProfiler. **k** A table of the known autophagy receptors and those detected (check marks) in proteomic analysis of the enriched proteins in *Atg7* cKO and *Atg14* cKO mouse brains (*p* < 0.05, Log₂FC > 2 SD). The list of autophagy receptors was manually collated through literature searches. Quantification of the change of the indicated ER and SV proteins from *Atg7* cKO (**l**) and *Atg14* cKO (**m**) mouse brains vs. Control. *n* = 3 biologically independent replicates. Relative protein levels were normalized to β-actin. The controls' means were set to 1. Two-sided unpaired *t*-test. NS, no significance. All data are shown as mean ± SEM. **n** Electron microscopy (EM) images of Purkinje axons at the deep cerebellar nuclei (DCN) area from *Atg7*[f/f] and *Atg7*[f/f]-Pcp2-Cre mice. Scale bar, 1 μm.

## Loss of neuronal autophagy causes an aberrant increase of RI subunits and impairment of c-FOS activity in mouse brains

We next asked if autophagy-deficiency in neurons affects PKA subunit degradation and PKA activity in vivo. Immunoblots demonstrated that AKAP11, RIα, and PKA-Cα subunit levels are significantly increased in

*Atg7* cKO and *Atg14* cKO mouse brains (Fig. 6a, b). To explore the brain region specificity of the increased AKAP11 and RIα expression, we performed IF staining for AKAP11 and RIα in *Atg7* cKO and control brains. The overall fluorescence intensities of AKAP11 and RIα staining were markedly increased across many brain regions of the mutant

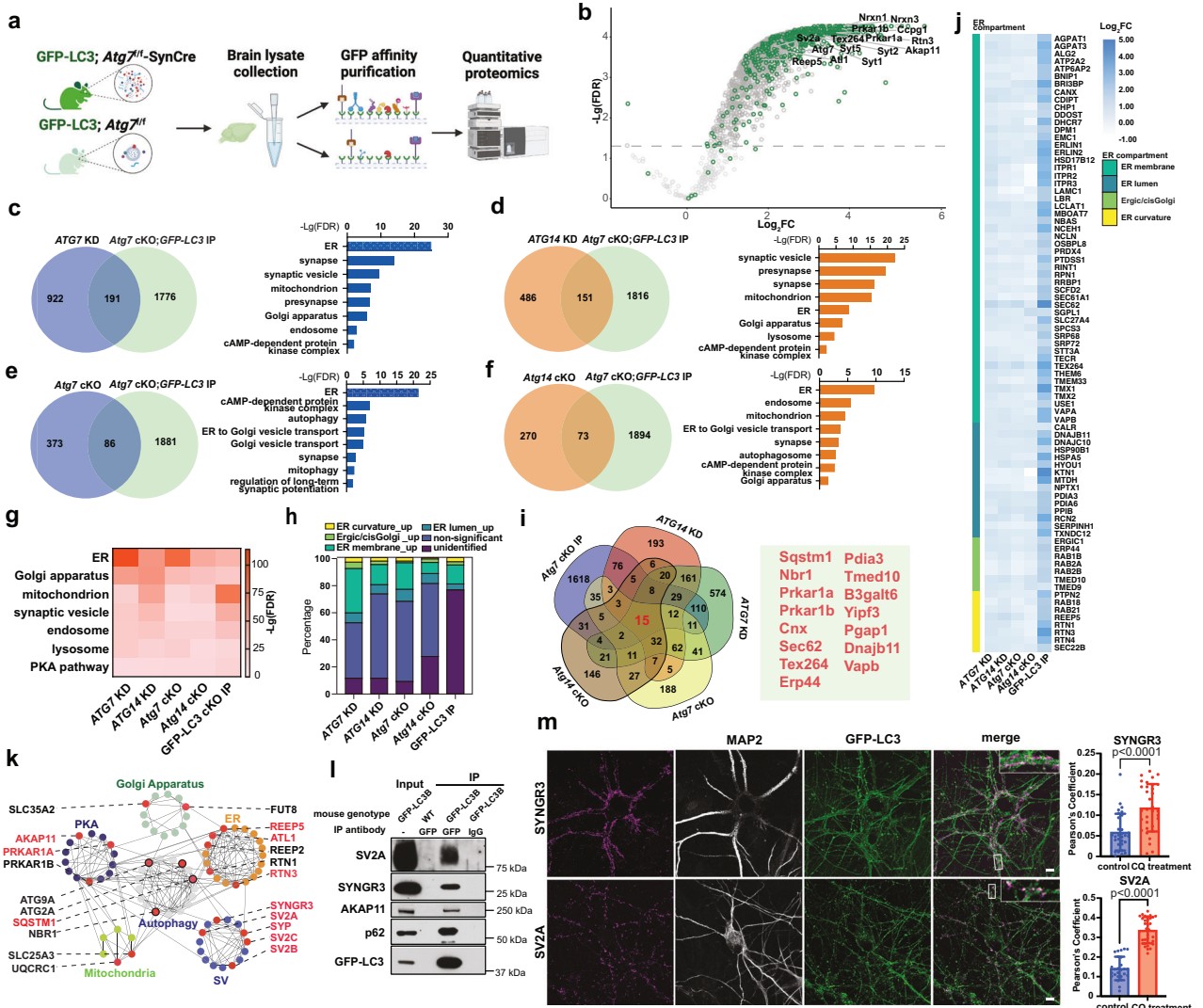

**Fig. 4 | Proteomic analysis of GFP-LC3-interacting proteins from neuron-specific *Atg7* cKO mouse brains. a** Schematic of LC3-interactome by combining genetic models and proteomics. Created with BioRender.com. **b** Volcano plot of proteomic results from DEPs between *Atg7*^f/f^-SynCre; GFP-LC3 IP vs. control IP (gray dots) and DEPs between *Atg7*^f/f^-SynCre; GFP-LC3 IP and *Atg7*^f/f^; GFP-LC3 IP (green dots). Threshold for enriched DEPs is $p < 0.05$ and Log$_2$FC > 2 SD (SD = 0.21). Dashed line indicates -Lg (*p*-value) = 1.3. Venn diagram (left) of overlapping DEPs between *ATG7* KD iNeurons (**c**), *ATG14* KD iNeurons (**d**), *Atg7* cKO mouse brain (**e**) or *Atg14* cKO mouse brain (**f**) and *Atg7*^f/f^-SynCre; GFP-LC3 IP. Bar graph displays the enriched GO annotations from the overlapping proteins. **g** Heatmap of GO annotations enriched in *ATG7* KD iNeurons, *ATG14* KD iNeurons, *Atg7* cKO mice, *Atg14* cKO mice and *Atg7*^f/f^-SynCre; GFP-LC3 IP. **h** Percentage of identified ER proteins and ER sub-compartment proteins in different generic models. Unidentified: not detected in the dataset. **i** Venn diagram of upregulated DEPs enriched in *ATG7* KD iNeurons, *ATG14* KD iNeurons, *Atg7* cKO mice, *Atg14* cKO mice and *Atg7*^f/f^-SynCre;

GFP-LC3 IP. Numbers indicate protein numbers. 15 proteins shared by all the datasets were shown in the right panel. **j** Heatmap shows Log$_2$FC of ER compartment proteins that are identified in all 5 datasets. **k** Six examples of enriched PPI modules in upregulated DEPs shared by *ATG7* KD and *ATG*14 KD iNeurons with the proteins validated in this study highlighted in red. Each dot represents a protein, and the interaction is indicated by connected lines. **l** Co-immunoprecipitation and immunoblot analysis through GFP-LC3 affinity pull-down from GFP-LC3 transgenic and WT mouse brains. **m** Immunofluorescence images (left) of GFP-LC3 knock-in human iNeurons with anti-SV proteins (SYNGR3 or SV2A). The cells were pretreated with chloroquine (CQ, 100 µM) for 24 h before fixation. Insets, magnified images of the boxed regions. Scale bar, 10 µm. Pearson's coefficient analysis of GLP-LC3 with SYNGR3 (n = 33 CQ-treated and 30 control iNeurons) or SV2A (n = 31 CQ-treated 28 control iNeurons) from 3 biologically independent replicates, two-sided unpaired *t*-test. All data are shown as mean ± SEM.

mice. For example, we found the accumulation of AKAP11 and RIα-positive cytosolic aggregate-like structures in the hippocampus (CA3 and DG), cortex (layer IV), amygdaloid, and thalamic nucleus (Fig. 6c–h). Additionally, we found AKAP11 colocalized with p62 in the cytosolic aggregates in neurons (Fig. 6e, f). IF staining of the mutant mouse brains revealed a reduced number of neurons labeled with c-FOS, indicating a decrease in neuronal activity at the basal level (Fig. 6i, j). Together, our work demonstrates that basal autophagy plays a critical role in maintaining the homeostasis of PKA-RI complex in neurons; loss of autophagy disrupts PKA signaling and impairs c-FOS mediated neuronal activity in vivo.

## AKAP11 mediates the degradation of PKA-RI protein complex in iNeuron

Finally, we asked whether AKAP11 plays a role in mediating the degradation of PKA RI (α and β) and PKA-Cα in neurons. We knocked down *AKAP11* gene expression in iNeurons using CRISPR interference technology and found a significant increase of PKA RIα, RIβ, and Cα subunit levels in two independent lines of iNeurons through immunoblotting. In contrast, the protein level of PKA RII subunit remains unaltered (Fig. 7a, b). The IF staining intensity for both RIα and RIβ proteins was markedly enhanced in the soma and the neuritic processes of *AKAP11* KD iNeurons, compared to the control iNeurons

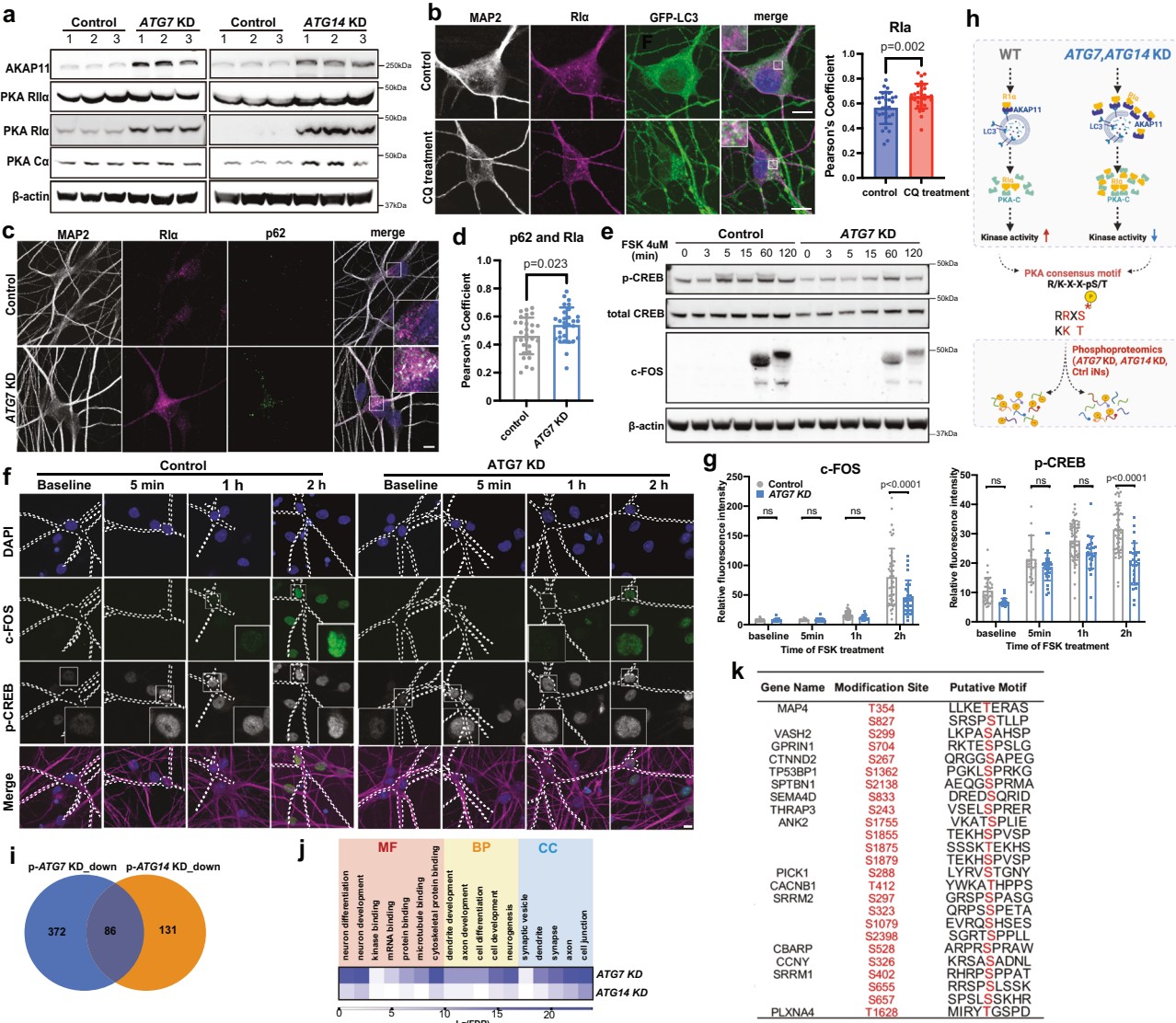

**Fig. 5 | Characterization of AKAP11, PKA subunit levels and activity, and C-FOS level in human iNeurons with *ATG7* KD or *ATG14* KD. a** Immunoblot analysis of AKAP11 and PKA subunits in human iNeurons. $n = 3$ independent biological replicates. **b** Immunofluorescence imaging of PKA-RIα, GFP-LC3 and MAP2 in human iNeurons expressing GFP-LC3. The cells were pretreated with chloroquine (100 μM) for 24 h to block autophagy. Insets, magnified images of the boxed regions. Scale bar, 10 μm. Pearson's coefficient of GFP-LC3 and PKA-RIα in control ($n = 31$ iNeurons) and chloroquine ($n = 30$ iNeurons) treatment from 3 biologically independent replicates, two-sided unpaired *t*-test. **c** Immunofluorescence images of PKA-RIα, p62, and MAP2 in control and *ATG7* KD iNeurons. Insets, magnified images of the boxed regions. Scale bar, 10 μm. **d** Pearson's coefficient of RIα and p62 in control ($n = 30$) and *ATG7* KD ($n = 30$) iNeurons from 3 biologically independent replicates, two-sided unpaired *t*-test. **e** Immunoblot analysis in control and *ATG7* KD iNeurons from i3N-derived iNeurons treated with Forskolin (4 μM) for 3, 5, 15, 60,

120 minutes. **f**, **g** Immunofluorescence images of control and *ATG7* KD ($n = 33, 26, 63, 49$ control iNeurons, $n = 25, 30, 27, 28$ *ATG7* KD iNeurons for baseline, 5 min, 1 h, and 2 h, respectively) human iNeurons treated with Forskolin (4 μM) for 0, 5, 60, 120 minutes. The total signal intensity of c-FOS and p-CREB in MAP2-positive iNeurons were quantified. Scale bar, 10 μm. Three biological replicates. Two-sided unpaired t-test. Ns, no significance (**g**). **h** Hypothetical changes of PKA subunit levels and activity and consequent alteration in the phosphorylation of PKA substrates in autophagy-deficient cells. Created with BioRender.com. **i** Venn diagram shows significantly downregulated phosphorylated peptides in *ATG7* KD and *ATG14* KD iNeurons (FDR < 0.1, $\log_2$FC < 0). **j** Heatmap shows enriched GO pathways for significantly downregulated phosphorylated peptides overlapped between *ATG7* KD and *ATG14* KD iNeurons using the dataset from **i**. CC Cellular Component, BP Biological Process, MF Molecular Function. **k** List of putative PKA substrates identified from the overlapped dataset in **i**. All data are shown as mean ± SEM.

(Fig. 7c, d). Thus, the above data demonstrates a critical role for *AKAP11* in mediating selective degradation of PKA-RI complex through autophagy in human neurons. Such a remarkable increase of the RIα and RIβ levels may have a profound impact on PKA-mediated signaling and neuronal activity.

## Discussion

Our integrated proteomic profiling and multidisciplinary experimental validation have delineated the landscape of autophagy degradation in human and mouse neurons. We show that basal autophagy targets

various cellular pathways and organelles, such as ER, mitochondria, Golgi, SV proteins, endosome, and cAMP-PKA pathway, for degradation in neurons. We find a function of neuronal autophagy in maintaining the homeostasis of PKA-RI complex through the autophagy receptor AKAP11 and regulating cAMP/PKA activity. Thus, our study offers a valuable resource for understanding the physiological function of autophagy in human neurons.

Previous investigations of neuronal autophagy cargo have provided useful information for the understanding of autophagy functions in neuron[10,25,27,68]. Our integrated proteomic analysis with

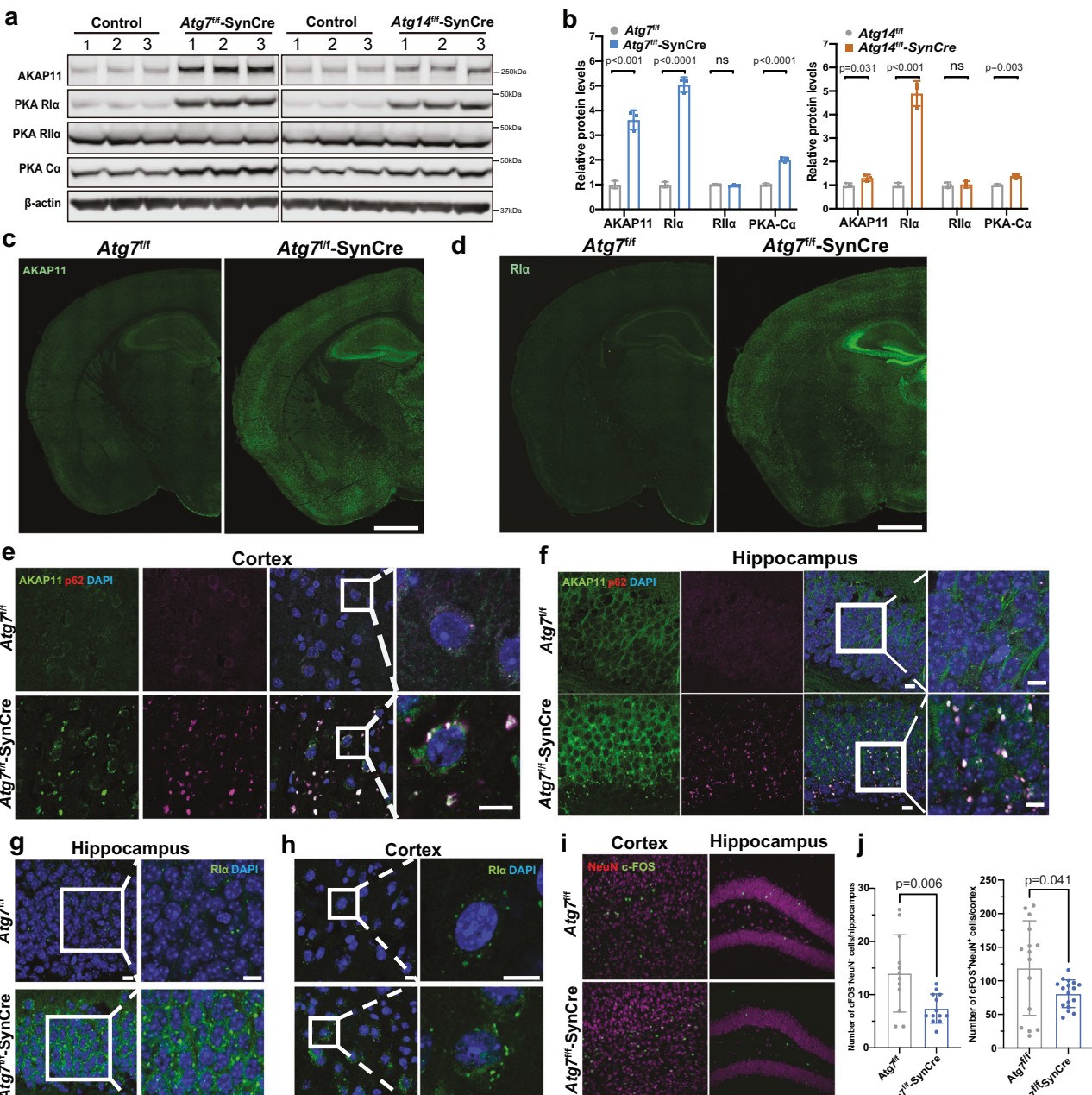

**Fig. 6 | Examination of AKAP11, PKA RI subunit, and c-FOS levels in *Atg7* cKO mouse brains. a** Immunoblot analysis of AKAP11 and PKA subunits as indicated in the whole brain lysates of *Atg7*f/f, *Atg7*f/f-Syn-Cre (left), and *Atg14*f/f, and *Atg14*f/f-Syn-Cre (right) mice (2-month-old), *n* = 3. **b** Quantification of the change of proteins from **a**. Relative protein levels were normalized to the loading control β-actin. The controls' means were set to 1. Two-sided unpaired *t*-test. Immunofluorescence images of *Atg7*f/f and *Atg7*f/f-SynCre mouse brain slices (2-month-old) stained with anti-AKAP11 (**c**) and anti-RIα (**d**) antibodies. *n* = 3, scale bars, 200 μm. Immunofluorescence images of *Atg7*f/f and *Atg7*f/f-SynCre mouse brain slices (2-month-old) co-stained with anti-AKAP11 (green) and anti-p62 antibodies (magenta) in the cortex (**e**) and hippocampus (**f**). *n* = 3, Scale bars, 10 μm. Immunofluorescence images of *Atg7*f/f and *Atg7*f/f-SynCre mouse brain slices (2-month-old) stained with anti-RIα (green) antibody in the hippocampus (**g**) and cortex (**h**). *n* = 3, Scale bars, 10 μm. **i** Immunofluorescence images of *Atg7*f/f and *Atg7*f/f-SynCre mouse brain slices (2-month-old) co-stained with anti-c-FOS and anti-NeuN antibodies in the cortex and hippocampus (DG) at the basal level. *n* = 3 for each genotype, 3–5 slides for each brain region. Scale bars, 100 μm. **j** Quantification of c-FOS⁺NeuN⁺ neurons in **i**. *n* = 3 mice. Two-sided unpaired *t*-test. All data are shown as mean ± SEM.

multiple mutant iNeuron lines and mice expands the investigations by providing insight on the neuronal autophagy functions. One key finding is the remarkable abundance of proteins from ER subcompartments[53] resulted from autophagy block, suggesting that ER is a most prominent target of neuronal autophagy. The significance of ER-phagy in neurons has been proposed in previous studies by using *Atg5* cKO mice[27] and human *ATG12*-deficient, NGN2-induced neuron[68]. Supporting the robustness of our approach for autophagy cargo identification, our results uncovered all six previously reported ER-

phagy receptors, including TEX264, FAM134B, RTN3L, SEC62, CCPG1, and ATL3, suggesting that these receptors are themselves the substrates of autophagy. Future study of the many ER protein cargoes identified from our study will assist in understanding the incredible complexity and selectivity of ER-phagy particularly in neurons[69].

Our results demonstrate that autophagy plays a role in maintaining the homeostasis of many SV trafficking proteins in neurons. Our data suggests that a set of SV-related proteins, including synaptogyrin1 (SYNGR1), synaptogyrin3 (SYNGR3), synaptophysin1 (SYP1),

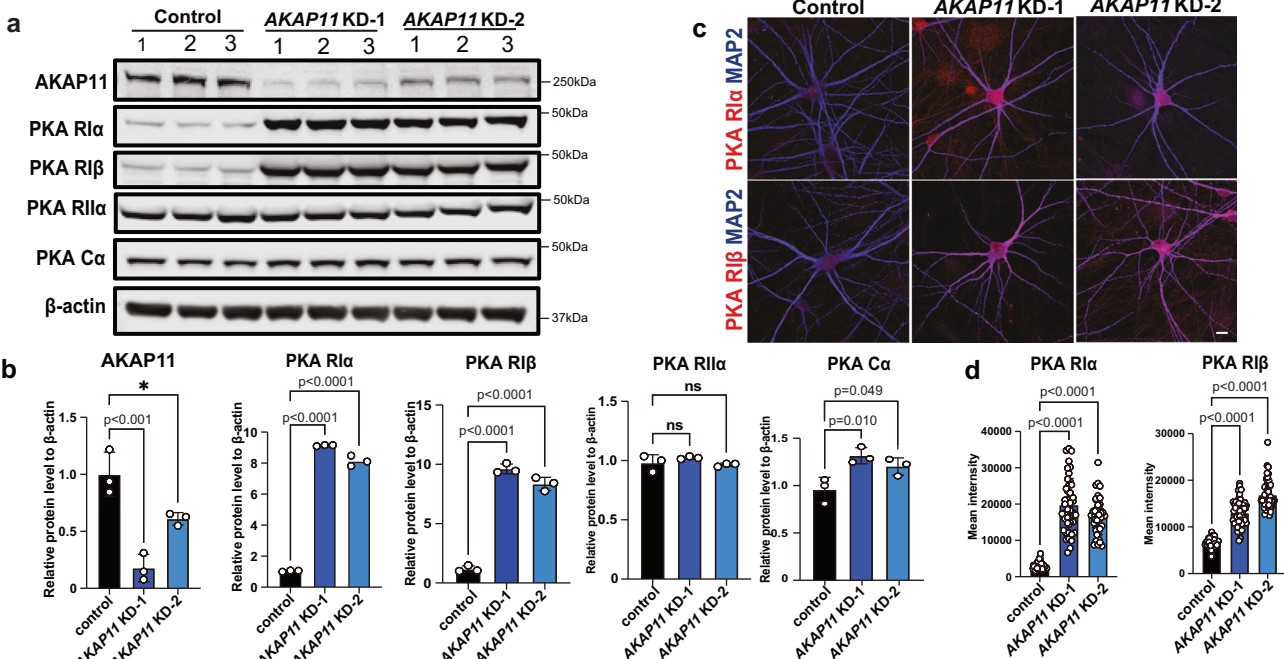

**Fig. 7 | Examination of protein levels of PKA subunits in human AKAP11 deficient iNeurons. a** Immunoblot analysis of AKAP11, and PKA subunits in the control and two clones of *AKAP11* KD (KD-1 and KD-2) human glutamatergic iNeurons. $n = 3$. **b** Quantification of the change of the indicated proteins from *AKAP11* KD iNeurons vs. Control. $n = 3$ biologically independent replicates for each protein. Relative protein levels were normalized to the loading control β-actin. The controls' means were set to 1. Two-sided unpaired *t*-test. NS, no significance. **c** Immunofluorescence images of control and two clones of *AKAP11* KD iNeurons co-stained with anti-PKA RIα/ RIβ (red) and MAP2 (Blue). $n = 3$, Scale bars, 10 μm. **d** The total signal intensity of PKA RIα ($n = 35, 45, 41$ control, *AKAP11* KD-1, *AKAP11*-KD-2 iNeurons) and RIβ ($n = 40, 53, 47$ control, *AKAP11* KD-1, *AKAP11*-KD-2 iNeurons) iNeurons were quantified. Data were collected from 3 biologically independent replicates. One-way ANOVA, Dunnett's multiple comparisons test for comparison between groups. Ns, no significance. All data are shown as mean ± SEM.

SV2A, and SV2C are potential autophagy targets in neurons. Although it was proposed that autophagy may recycle SV at presynaptic terminals[15], one study showed that disruption of autophagy has little impact on the number of SV. Our EM study of *ATG7* KD iNeuron also suggests no significant change of SV number (Supplementary Fig. 3L), arguing that autophagy may not directly turn over SV per se at the synapse[27]. Nonetheless, our observation may provide insight into the mechanism for how autophagy regulates presynaptic activity[14].

Another significant finding in our study is the identification of the specific PKA-RI complex as a target of autophagy in neurons under basal condition. Our results demonstrate that neuronal autophagy constantly degrades RI regulatory (α/β) and Cα catalytic subunits of PKA, therefore controlling PKA homeostatic activity. A recent study reported that autophagy regulates cAMP/PKA signaling pathway in neuron specific *Atg5* KO mouse excitatory neurons[70], corroborating our observations. Our study further showed a function of AKAP11 as an autophagy receptor in selective degradation of PKA RI regulatory (α/β) and Cα catalytic subunits in neurons, a function likely conserved in different cell types as we previously showed[57]. We observe discrete AKAP11 and RIα puncta, which colocalize with p62, in neurons at multiple brain regions, such as the CA3 and DG regions of the hippocampus, layer IV of the cortex, amygdaloid, and thalamic nucleus from *Atg7* cKO mice, suggesting that p62 may act as a co-receptor for AKAP11-mediated degradation, and together, they regulate PKA activity in the brain. Previous studies have shown that PKA regulates neuronal signaling pathways which is essential for learning and memory in mice and flies[61,62]. Our findings thus provide additional insight into the mechanism whereby neuronal autophagy control synaptic activity underlying learning and memory[71].

Our study shed light on the molecular mechanism underlying multiple neurological disorders. A recent study reported 12 individuals from 5 independent families that carried biallelic, recessive variants in

human *ATG7* associated with severe reduction or complete loss of ATG7 expression. These individuals suffered complex neurodevelopmental disorders[16]. Our systematic study, particularly with human iNeurons carrying *ATG7* deletion, provides a comprehensive view for what cellular pathways and functions could be disrupted in the patients' neurons carrying *ATG7* mutations. In Alzheimer's disease (AD), dystrophic neurites and amyloid plaques are the important hallmarks of neuropathologies in the affected brains[72]. Morphological and molecular analysis of AD brains and mouse models showed a sequential and distinct enhancement of staining of ATG9, RTN3, REEP5, RAB7, LC3, and LAMP1 proteins during dystrophic neurite and amyloid plaque development[73]. The increased ATG9, RTN3 and REEP5 protein levels in AD neurites surrounds the early-stage amyloid plaques, suggesting that a block of ER-phagy occurs early and may contribute to the formation of dystrophic neurites and amyloid plaques in affected AD brains. Furthermore, rare *AKAP11* truncating variants were recently identified as significant risk factors for both bipolar disorder and schizophrenia from exome sequencing[74]. Our observation raises a possibility that the pathogenic mechanism of the psychiatric disease involves disruption of PKA-RI complex homeostasis and cAMP-PKA signaling in neurons.

We also recognized the limitations of our study. For example, the long-term inhibition of autophagy function through deletion of the essential autophagy genes may cause compensatory responses with specific changes in certain accumulated proteins resulting from disturbing cellular homeostatic mechanisms, such as ER stress, post-transcriptional modifications, and protein synthesis, etc[75]. Our approaches, however, is unable to exclude those proteins with increased levels caused potentially by the above conditions. Furthermore, our study has yet to examine neuronal interactome of additional ATG8 homologs (e.g., GABARAP, GABARAPL1/2), despite our analysis of the LC3-interactome. Therefore, our studies await

future thorough validation of the putative novel autophagy cargo in neurons.

In summary, our comprehensive approach of deep proteomic profiling provides a global view of autophagy degradation in human and mouse neurons, offers a valuable resource of neuronal autophagy research, and shed light on the molecular mechanisms of several neurological disorders.

## Methods

### Animals

All animal experiments were approved by the Icahn School of Medicine at Mount Sinai Institutional Animal Care and Use Committee (IACUC-2015-0046) and were conducted in compliance with the relevant ethical regulations. Mice were maintained in social cages on a 12 h light/dark cycle with free access to food and water; animals were monitored daily for food and water intake.

Floxed *Atg7* (*Atg7*flox/flox) mice were kindly gifted from Dr. Masaaki Komatsu (Tokyo, Japan)[28,76]. *Atg14*flox/flox mice were from Dr. Herbert W. Virgin (Washington University School of Medicine, St. Louis, MO)[77,78]. GFP–LC3 transgenic mice (C57BL/6J) were described previously[55]. Synapsin-Cre mice were purchased from Jackson Lab (#003966). *Atg7*f/f and *Atg14*f/f mice were crossed with *Atg7*f/+-SynCre and *Atg14*f/+-SynCre mice to generate *Atg7*f/f-SynCre and *Atg14*f/f-SynCre mice. *GFP-LC3*; *Atg7*f/+-Syn-Cre mice were crossed with *Atg7*f/f mice to generate *GFP-LC3*; *Atg7*f/f-SynCre mice. For obtaining *Atg7*-deficient cultured primary neurons (postnatal day 0), *Atg7*flox/flox mice were bred to *Atg7*flox/+;Synapsin-Cre mice. To generate the *Atg7*flox/flox-PCP2-Cre mice, *Atg7*flox/+ mice were crossed with Pcp2-Cre mice to generate *Atg7*flox/+-PCP2-Cre mice. *Atg7*flox/flox mice were crossed with *Atg7*flox/+-PCP2-Cre mice to generate *Atg7*flox/flox-PCP2-Cre mice, in which *Atg7* expression is specifically deleted in cerebellar Purkinje cells (Pcp2-Cre). Animals of both sexes were used in the analyses.

### Cell culture

All human ESCs and PSCs were maintained on Geltrex-coated plates in feeder-free Stemflex medium and a 5% $CO_2$ environment at 37 °C. Cells were passaged using 0.5 mM EDTA/PBS in Stemflex supplemented with 2 mM Thiazovivin or 50 nM Chroman1. Thiazovivin/Chroman1 was removed from the medium on the following day. Research performed on samples of human origin was conducted according to protocols approved by the Institutional Review Boards of Icahn School of Medicine at Mount Sinai. H1 (WA01) ES cells (from a male parental cell line) were obtained from WiCell Research Resources (#Ae001-A); The inducible i3N iPS cell line (from the WTC11 iPSC parental cell line, male) was kindly provided by Dr. Martin Kampmann's lab[34]. Mono-ALLELIC mEGFP-Tagged MAP1LC3B WTC iPSC Line were purchased from Coriell Institute (#AICS-0030-022) which is derived from a male parental iPS cell line, human embryonic kidney (HEK) 293 T/17 cells were purchased from ATCC (CRL-11268).

HEK293T cells were maintained in MEF medium (Dulbecco's modified Eagle's medium (Gibco, 11965-092) supplemented with 10% Cosmic Calf Serum (HyClone, SH30087.03), 1× NEAA (Gibco, 11140-050), 1×Sodium Pyruvate (Gibco, 11360-070), 0.008% β-mercaptoethanol).

Mouse glial cultures were generated from cortical hemispheres at postnatal day 3 (P3). Briefly, the cortices from 3 pups were incubated in 5 mL of 20 Units/mL Papain, 0.5 mM EDTA, and 1 mM $CaCl_2$ in HBSS for 15 minutes. After incubation, the tissues were manually dissociated by forceful trituration. Cells were filtered using a 70 μm cell strainer. The resulting cells were grown in MEF medium at 37 °C with 5% $CO_2$.

### sgRNA design and cloning to knock down *ATG7* and *ATG14* using CRISPR interference

4 small guide RNA (sgRNA) candidates for *ATG7* and 4 sgRNA candidates for *ATG14* were designed using the Broad Institute CRISpick web-based tool (https://portals.broadinstitute.org/gppx/crispick/public).

sgRNAs were cloned into the Lenti-U6-dcas9-krab-Puro (Addgene #71236) plasmid using the Golden Gate DNA Assembly kit (NEB, E1602S). sgRNAs sequences were confirmed by Sanger sequencing. WT human H1 stem cells were transduced with lentiviruses over-expressing sgRNA-dcas9-krab-puro and puromycin (1 μg/ml) was applied for at least 3 days to enrich puromycin resistant cells. Western blotting was performed to confirm the efficiency of *ATG7* and *ATG14* knock down. *ATG7* sgRNA#3 and *ATG14* sgRNA#3 were confirmed to have the highest knock down efficiency. SgRNAs cloning were used th following oligo pairs (IDT):

ATG7 sgRNA#1, Forward oligo: 5'-CACCGAGGACCGCGTTGCGT-CATCG-3', reverse oligo: 5'-AAACCGATGACGCAACGCGGTCCTC-3',

ATG7 sgRNA#2, Forward oligo: 5'-CACCGACTTACCGCCGCT-CAACTTC-3', reverse oligo: 5'-AAACGAAGTTGAGCGGCGGTAAGTC-3',

ATG7 sgRNA#3, Forward oligo: 5'-CACCGGGCGGTAAGTGAGCC GCGGC-3', reverse oligo: 5'-AAACGCCGCGGCTCACTTACCGCCC-3',

ATG7 sgRNA#4, Forward oligo: 5'-CACCGTCCCAGTGGCAAGC GCGGGC-3', reverse oligo: 5'-AAACGCCCGCGCTTGCCACTGGGAC-3',

ATG14 sgRNA#1, Forward oligo: 5'-CACCGGGCGATTTCGTC-TACTTCGA-3', reverse oligo: 5'-AAACTCGAAGTAGACGAAATCG CCC-3',

ATG14 sgRNA#2, Forward oligo: 5'-CACCGATCGCCGCTCT-GAACGCATT-3', reverse oligo: 5'-AAACAATGCGTTCAGAGCGGC GATC-3',

ATG14 sgRNA#3, Forward oligo: 5'-CACCGACTCCGTGGAC-GATGCGGAG-3', reverse oligo: 5'-AAACCTCCGCATCGTCCACGGAG TC-3',

ATG14 sgRNA#4, Forward oligo: 5'-CACCGCCCACTGGGAGAC GCCATGA-3', reverse oligo: 5'-AAACTCATGGCGTCTCCCAGTGGGC-3',

NTC sgRNA, Forward oligo: 5'-CACCGTATTACTGATATTGGTGGG-3', reverse oligo: 5'-AAACCCCACCAATATCAGTAATAC-3'.

### Cell line generation

**H1-Lenti-*ATG*7/14 KD cell line.** Human H1 stem cells were dissociated using 0.5 mM EDTA/PBS and plated in Stemflex supplemented with 50 nM Chroman1. Lentiviruses (Addgene #71236) expressing *ATG7* sgRNA #3, *ATG14* sgRNA #3, or an empty vector were used to transduce H1 cells during passaging. The medium was changed to Stemflex supplemented with puromycin (1 μg/ml) for 3 consecutive days. Puromycin resistant cells were dissociated using Accutase for 15 min and seed at a density of 2000 cells per 10 cm dish. Individual clones were harvested and expanded. Protein expression was confirmed by immunoblotting assay. *ATG7* KD clone #1 and #6, *ATG14* KD clones #3 and #5, and empty vector clones #2 and # 6 were confirmed by western blotting and used for subsequent experiments.

**H1-dcas9-krab cell line generation.** An FRT-EMCV-IRES-Neo-FRT-EF1a-dcas9-HA-krab transgene was subcloned into a pSIN donor vector containing an AAVS1 homology arm using CRISPR Cas9. Genomic DNA from Neomycin-selected cells was expanded, purified, and genotyped by three PCR reactions. The following PCR primers (IDT) were used for the genotyping of cell line generation:

5'GT, 5'-ACTTTGAGCTCTACTGGCTTCTG-3'(forward),
5'GT, 5'-CGAAGTTATGTTAACGAAGTTCCTATAC-3'(reverse),
3'GT, 5'-GACAATAGCAGGCAATAACTTCG-3'(forward),
3'GT, 5'-GAACGGGGCTCAGTCTGA-3'(reverse),
WT, 5'-CTGTCATGGCATCTTCCAGG-3'(forward),
WT, 5'-GATCAGTGAAACGCACCAGAC-3'(reverse).

All the cell lines generated in this paper were karyotyped normal.

### Fluorescence microscopy of ER-phagy reporter

WT and *ATG7* KD human iNeurons were cultured on coverslips coated with GelTrex. RFP-GFP-KDEL (Addgene, #128257) plasmids were delivered to WT and *ATG7* KD human iNeurons via jetOPTIMUS transfection reagent (Polyplus, #02BIM1804B5). 48 h post

transfection, the medium was changed to Glucose and Sodium pyruvate-free NBA (Gibco, A24775-01) supplemented with B27 (Life Technologies, 17504044), GlutaMAX (Gibco, 35050061), 1% dialyzed Fetal Bovine Serum (Gibco, A3382001) for 48 h and fixed with 4% PFA at room temperature for 10 min and washed with PBS buffer for 3 times. Cells were mounted with Antifade Mounting Medium with DAPI (Vector Laboratories, Cat# NC9524612, 0.5 µg/ml) directly then imaged by fluorescence microscopy and processed in Fiji (ImageJ) and Adobe Photoshop.

## Lentivirus production

Lentiviruses were produced in HEK293T cells (passage number lower than 20) by co-transfection with three helper plasmids (pRSV-REV, pMDLg/pRRE, and vesicular stomatitis virus G protein expression vector) using Polyethylenimine (PEI, Polysciences, #23966-1)[79]. The following plasmids were used in the lentivirus packing: Tet-O-Ngn2-puromycin[33], FUW-M2rtTA[80], pRSV-rev (Addgene, #12253), pMDLg (Addgene, #12251), pCAG-VSVG (Addgene, #64084). To be brief, transfection mixtures A and B (one per lentivirus) were prepared as following (per each 10 cm): Tube A: 500 µl DMEM + 20 µg total DNA (e.g. 10 µg TetO-FUW-GFP or any other lentivirus plasmid, 5 µg PMDL, 2.5 µg RSV, 2.5 µg VSVg). Tube A and B were combined at the maximum speed for 10 seconds by vortex. Tube B: 500 µl DMEM + 60 µl 1X PEI, mix by vortexing. The mixture were incubated at room temperature for 20 min and vortexed one more time and added dropwise to the cells (1 ml per plate). At this point, cells were 90% confluent. 6 h later, cells were washed with PBS and 6 ml of fresh prewarmed MEF media was added to each plate. 30 h after the last media exchange, GFP expression was checked. The first media collection was only proceed if GFP is expressed by >than 70% of the cells. 46 h post-transfection, a second collection was performed, and the plates were discarded. The I and II collections were combined. Lentiviral particles were ultracentrifuged at 75,300 × g, 4 °C for 2 h, re-suspended in DMEM, aliquoted, and stored at −80 °C. Only virus preparations with >90% infection efficiency as assessed by EGFP expression or puromycin resistance were used for experiments.

## Generation of neurons from human PSCs

Glutamatergic neurons were generated by overexpression of the transcription factor Ngn2 as previously described[33]. Briefly, human PSCs were dissociated using 0.5 mM EDTA/PBS and plated at a density of 88,000 cells/cm² in N2 medium supplemented with 50 nM Chroman1. At the same time, cells were mixed with FUW-TetO-Ngn2-P2A-puromycin and FUW-rtTA lentiviruses by adding them directly to the medium. 24 h later, the medium was replaced by N2 medium (1×N2 supplement, 1×NEAA in DMEM-F12 medium) containing Doxycycline (2 mg/mL) to induce transgene expression. Transduced cells were enriched with puromycin (1 mg/mL) for 2 days. 4-5 days post Doxycycline, neurons were dissociated and plated together with mouse glial cells (100,000 cells/cm²) on Geltrex-coated plates. Two weeks after transgene induction, Doxycycline was removed and the neuronal culture was maintained in Neurobasal A medium supplemented with 1× B27, 1× Glutamax, and 1% fetal bovine serum. Mature neurons were used for various experiments on day 42 (after 5 weeks of co-culture with mouse glia).

## RNA extraction and mRNA sequencing

To determine gene expression level, ATG7 KD and Control iNeurons (Day 42, 1 well of a 6-well plate around 2.5 × 10⁵ cells total) were washed with PBS and lysed in 1 mL of Trizol for RNA extraction. Following lysis, DNA contamination was removed using TURBO DNA-free kit treatment. 3 biological replicates were obtained for both ATG7 KD and Control iNeurons.

Three pairs of mouse whole brain tissue from Atg7^{f/f} and Atg7^{f/f}-SynCre mice (6-8 weeks old, both male and female were included)

were dissected and snapped frozen using liquid nitrogen. Mice whole brains were ground using the Mortar & Pestle Set (Thermofisher, Cat# FB961N and 50-195-4054054).

## Western blot analysis of cell lysate

Cultured cells were harvested and subjected to western blot. Cells were lysed in lysis buffer (1% Triton-X 100, 50 mM Tris HCl (pH = 7.5), 150 mM NaCl, proteinase/phosphatase inhibitor, and EDTA) and supernatants were collected on ice. Protein concentrations were measured by the Pierce BCA Protein Assay Kit in accordance with the manufacturer's protocol. 25 µg of total proteins were run on 4%-12% 15-well Bis-Tris gel and proteins were transferred to a PVDF membrane. The resulting membranes were blocked in a blocking buffer (5% non-fat milk in TBST or LI-COR Blocking Buffer (LI-COR, 927-60001) buffer for 1 hour at room temperature. Primary antibodies were diluted in blocking buffer as described above and incubated at 4 °C for 24 h. Membranes were incubated with primary antibodies, including SEC62 (Abcam, #ab140644, 1:1000), ATL1 (Cell Signaling Technology, #12728, 1:1000), RTN3 (Proteintech, # 12055-2-AP, 1:1000), REEP5 (Proteintech, #14643, 1:5000), Calnexin (Cell Signaling Technology, #2679 P, 1:1000), TEX264 (NOVUS biologicals, #NBP1-89866, 1:1000), SV2A (SYSY, #119022, 1:1000), SV2B (SYSY, #119102, 1:1000), SV2C (SYSY, #119202, 1:1000), SYNGR3 (santa cruz, # sc-271046, 1:5000), SYNGR1 (SYSY, #103002, 1:1000), SYP (SYSY, #101011, 1:1000), synapsin1 (synaptic systems, #106004, 1:1000), AKAP11 (Life Span Bio Sciences, #LS-C374339-200, 1:1000), PKA RIα (Cell Signal, #5675, 1:1000), PKA RIIα (BD Bio Sciences, #612242, 1:1000), Cα (Cell Signal, #4782S, 1:1000), ubiquitin (DAKO, #Z0458, 1:1000), c-FOS (Abcam, #AB208942, 1:1000), c-FOS (SYSY, #226017, 1:1000), CREB (Abcam, #ab31387, 1:1000), p-CREB Ser133 (Cell Signal, #9198S, 1:1000), ATG7 (Cell Signaling Technology, #8558S, 1:1000), ATG14 (MBL, #PD026-006, 1:1000), p62 (MBL, #PM066, 1:1000), GAPDH (Invitrogen, #MA5-15738, 1:1000), β-actin (Cell Signaling Technology, #3700S, 1:1000), HSP90 (Cell Signaling, #4874S, 1:1000), LC3A/B (Cell Signaling Technology, #12741S, 1:1000), GFP (Abcam, #ab13970, 1:1000). Secondary antibodies were diluted in blocking buffer and incubated for 1 h at room temperature. Membranes were visualized and processed with Image Lab and Adobe Photoshop.

## Western blot analysis of mouse brain lysate

Whole brains were collected from mice described above at 6–8 weeks of age and homogenized with a lysis buffer containing 50 mM Tris HCl, pH 7.5, 150 mM NaCl, 1% Triton X-100, and proteinase/phosphatase inhibitor. Brain lysates were subjected to immunoblotting assay following the same protocol for cultured cells.

## Immunofluorescence staining in cultured cells

Coverslips were washed once with PBS and then fixed with 4% PFA at room temperature for 10 min and with methanol (prechilled in −20 °C) in −20 °C for 30 min. Cells were then permeabilized with 0.2% Triton-X-100 for 10 min. The cells were then blocked with blocking buffer (1% CCS, 4% BSA in PBS) for 1-2 h at room temperature. Primary antibodies, including Calnexin (Cell Signaling Technology, #2679P, 1:200), SV2A (SYSY, #119022, 1:500), SYNGR3 (santa cruz, # sc-271046, 1:200), SYNGR1 (SYSY, #103002, 1:1000), SYP (SYSY, #101011, 1:200), synapsin1 (synaptic systems, #106004, 1:500), PKA RIα (Cell Signal, #5675, 1:200), NeuN (Sigma-Aldrich, #MAB377, 1:1000), c-FOS (Abcam, #AB208942, 1:1000), c-FOS (SYSY, #226017, 1:1000), CREB (Abcam, #ab31387, 1:200), p-CREB Ser133 (Cell Signal, #9198S, 1:200), MAP2 (Abcam, #ab5392, 1:5000), p62 (MBL, #PM066, 1:1000), LC3B (Cell Signaling Technology, #3868S, 1:200), GFP (Abcam, #ab13970, 1:5000), diluted in blocking buffer containing 0.03% Triton-X-100 were incubated overnight at 4 °C. Secondary antibodies diluted in PBS containing 0.03% Triton-X-100 were incubated at room temperature for 1 h. Images were obtained using a confocal microscope with Zen

2011 software (Zeiss LSM 780, Carl Zeiss, Jena, Germany) at 40x and 63x and the figures were processed with Fiji (ImageJ) and Adobe Photoshop.

## Immunofluorescence staining in mice brains
At least 3 pairs of mouse brains (6–8 weeks old) were perfused with 4% PFA for 10 min. They were fixed overnight in 4% PFA and 30% sucrose for 2 days at 4 °C. After the removal of sucrose, brains were placed in OCT compounds for cryosection and then incubated overnight at −80 °C. The block was transferred to the cryostat 30 min before cutting and incubated, and then sections were cut into 30 m thick at −20 °C. These slices were transferred to 24-well plate in PBS. Tissues were then permeabilized with 0.3% Triton-X-100 for 20 minutes and blocked with blocking solution consisting of 5% goat serum and 0.3% Triton-X-100 for 1 h at room temperature. Primary antibodies were incubated overnight at 4 °C, including AKAP11 (Life Span Bio Sciences, #LS-C374339-200, 1:200), PKA RIα (Cell Signal, #5675, 1:200), NeuN (Sigma-Aldrich, #MAB377, 1:1000), c-FOS (Abcam, #AB208942, 1:1000), c-FOS (SYSY, #226017, 1:1000), and secondary antibodies were incubated at room temperature for 1 h. Imaging was then performed with confocal microscopy using Zen 2011 software (v2.6) on 10×, 20×, and 63× objectives using Z-stack and tile scan tools and analyzed with Fiji (image J).

## Forskolin treatment of human iNeurons
Human i3N PSC cell lines were transduced with Lenti-*ATG7*-hygromycon-BFP viruses to knock-down ATG7 and Lenti-NTC-hygromycin-BFP viruses as control. Cells were selected using hygromycin B (Life Technologies, #10843555001, 1:250) every two generations. PSC cells were plated in the 6-well plate in Stemflext medium supplemented with 50 nM Chroman1. 24 h later, the medium was replaced by N2 medium containing Doxycycline (2 mg/mL) to induce neuron differentiation. For immunofluorescence staining, 4-5 days post Doxycycline, neurons were dissociated and plated together with mouse glial cells (neuron density at 100,000 cells/cm$^2$) on Geltrex-coated plates. Two weeks after transgene induction, Doxycycline was removed and the neuronal culture was maintained in Neurobasal A medium supplemented with 1× B27, 1× Glutamax, and 1% fetal bovine serum. Mature neurons on day 42 (after 5 weeks of co-culture with mouse glia) were treated with 4 μm Forskolin (Sigma-Aldrich, # F3917) for 5 min, 1 h and 2 h. Neurons were fixed and subjected to immunofluorescence staining as described above. For western blot, PSC cells were plated in 6-well plates at density of 150 k/cm$^2$ in Stemflext medium supplemented with 50 nM Chroman1. 24 h later, the medium was replaced by N2 medium containing Doxycycline (2 mg/mL) to induce neuron differentiation and was changed to fresh N2 plus Doxycycline every 2 days. Medium was changed to Neurobasal medium supplemented with 1× B27, 1× Glutamax, and 1% fetal bovine serum until 14 days post Doxycycline treatment. Neurons on day 14 were treated with 4 μm Forskolin for 5 min, 1 h and 2 h. Cells were harvested for western blot as described above.

## Screening strategy for ideal candidate gene from autophagy-deficient human iNeurons MS data
A multi-step screening process was used to determine the most ideal candidate gene for our study. The first step in establishing a list of candidate genes was to screen for genes in the *ATG7* and *ATG14* KD human iNeurons proteomics data that had a Log$_2$FC greater than the standard deviation (Log$_2$FC > SD) and a Log$_2$FC greater than 0 (Log$_2$FC > 0). These genes were then subjected to GO:TERM analysis by a web-based tool to determine which genes were related to the ER. SynGo was performed in the webtool (https://www.syngoportal.org/).

## Electron microscopy (EM)
Three pairs of *Atg7*$^{f/f}$ and *Atg7*$^{f/f}$-PCP2-Cre mice (6-8 weeks old) were sacrificed, and samples were processed following the procedures.

Briefly, 3 pairs of *Atg7*$^{f/f}$ and *Atg7*$^{f/f}$-PCP2-Cre mice were perfused and fixed with 2% paraformaldehyde and 2% glutaraldehyde in 0.1 M PB, pH 7.4. 40 mm vibrotome-cut utra-thin (~70 nm) sections were collected on uncoated 200-mesh grids. Grids were viewed with a TecnaiSpiritBT Transmission Electron Microscope (FEI) at 80 KV and pictures were taken with Gatan 895 ULTRASCAN Digital Camera.

For morphology EM on cells, 3 pairs of independently prepared *ATG7* KD and control human iNeuron cells (42 days post differentiation) grown on Permanox slides (Electron Microscopy Sciences (EMS), Hatfield, PA) were taken from incubation, and directly placed in 2% glutaraldehyde and 2% paraformaldehyde /0.1 M sodium cacodylate buffer for a minimum of 2 h at 4 °C. Cells were washed, fixed in 1% aqueous osmium tetroxide at RT for 1 h, washed, and transferred to 2% aqueous uranyl acetate at RT for 1 h. Slides were washed with distilled water, dehydrated in an ascending aqueous ethanol series, and then embedded in Epon resin. Inverted BEEM capsules (#3) were placed directly over regions of interest, filled with fresh resin, and transferred to a vacuum oven for heat polymerization at 60 °C for 12–24 h. To separate the cells from the slides, a hot plate was heated to 60 °C, and the slides were placed directly on a pre-heated hot plate for exactly 3 min and 30 s. The capsules were removed from the hot plate and carefully dislodged free from the slide using a plier. Ultrathin (85 nm) sections were collected onto 300 mesh copper grids (EMS) using a Leica UC7 ultramicrotome (Leica Biosystems Inc., Buffalo Grove, IL), contrast stained with uranyl acetate and lead citrate, and imaged on a Hitachi 7700 transmission electron microscope (Hitachi High Technologies America, Inc., Dallas, TX) equipped with an AMT 2 K × 2 K digital camera (Advanced Microscopy Techniques, Corp., Woburn, MA).

## Sample preparation and quantitative proteomics
**Autophagy-deficient human iNeurons and mouse brains.** Three batches of H1-Lenti-*ATG7/14* KD derived Ngn2 iNeurons were included for the quantitative proteomics and phosphoproteomics analyses (day 42 post differentiation, two subclones for each genotype. Three samples of *ATG7* KD clone # 1 and clone #6, one sample of *ATG14* KD clone #3 and two samples of *ATG14* KD clone#5, three samples of Vector clone #2 and four samples of clone #6). The cells were washed with PBS once and manually dissociated by gentle trituration in PBS. Cells were then centrifuged at 500 g for 3 min. The supernatant was removed, and cell pellet was snap-frozen by liquid nitrogen and stored at −80 °C before being subject to proteomics analyses. For mouse brain samples, 6–8-week-old autophagy-deficient mice (3 *Atg7*$^{f/f}$-SynCre and 4 *Atg7*$^{f/f}$ mice; 3 *Atg14*$^{f/f}$-SynCre and 4 *Atg14*$^{f/f}$ mice) were sacrificed and whole brains were dissected, and snap frozen in liquid nitrogen and stored in −80 °C before proteomics analyses.

**Protein extraction, quantification of iNeurons and mouse brain samples.** iNeuron cell pellets and mouse brain samples were lysed, and the protein concentrations were quantified. Briefly, the frozen samples were homogenized in the lysis buffer (50 mM HEPES, pH 8.5, 8 M urea, and 0.5% sodium deoxycholate) with 1X PhosSTOP phosphatase inhibitor cocktail (Sigma-Aldrich). Protein concentration was measured by the BCA assay (Thermo Fisher, # 23225) and then confirmed by Coomassie-stained short SDS gels.

**Protein digestion and TMT labeling for iNeurons and mouse brain samples.** The analysis was performed with a previously optimized protocol[45] -0.1 mg of iNeurons protein samples and ~0.1 mg of mouse brain protein samples (in lysis buffer with 8 M urea) were first digested by Lys-C (Wako, 1:100 w/w) at 21 °C for 2 h and then diluted by 4-fold to reduce urea to 2 M by 50 mM HEPES followed by the addition of trypsin (Promega, 1:50 w/w) to continue the digestion at 21 °C overnight. The digestion was terminated by the addition of 1% trifluoroacetic acid. After centrifugation, the supernatant was desalted

and then dried by Speedvac. Each sample was resuspended in 50 mM HEPES (pH 8.5) for TMT labeling, and then mixed equally followed by desalting. The iNeuron samples were labeled by TMT 16. The *Atg7* cKO and *Atg14* cKO mouse brain samples were labeled by two sets of TMT 10 respectively, but only 7 TMT channels was discussed here. All TMT channel information can be found in Supplementary Data 1a.

### Phosphopeptide enrichment of iN samples
Phosphopeptide enrichment was carried out by TiO$_2$ beads as described previously[81]. Briefly, the pooled desalted TMT labeled peptides (-1.8 mg) were dissolved in the binding buffer (65% acetonitrile, 2% TFA, and 1 mM KH2PO4). TiO$_2$ beads (7.2 mg) were washed twice with the washing buffer (65% acetonitrile, 0.1% TFA), incubated with the peptide at 21 °C for 20 min. The TiO$_2$ beads were then washed twice again with washing buffer and loaded into a C18 StageTip (Thermo Fisher), followed by the elution of phosphopeptides by the basic pH buffer (15% NH4OH, and 40% acetonitrile). The eluates were dried respectively and dissolved in 5% formic acid for LC-MS/MS analysis. The flowthrough of phosphopeptide enrichment was desalted and dried for basic LC fractionation.

### GFP-LC3 affinity purification from mouse brains, in-gel digestion, and TMT labeling
The GFP-LC3 overexpression and control mouse brains (3 pairs of 6-8 weeks old mice, both male and female mice were used) were harvested and rinsed with cold PBS. Homogenization buffer A (0.32 M sucrose, 1 mM NaHCO3, 0.25 mM CaCl2, 1 mM MgCl2, 50 mM Tris HCl, pH 7.5) supplied with halt proteinase and phosphatase inhibitors (Invitrogen, #78442) was added as 2 ml buffer A per mouse brain. The tissue was homogenized with a glass douncer 20 times before being centrifuged at 1000 × *g* for 10 min. The supernatant was transferred to a new 15-ml falcon tube labeled as the cytoplasmic sample. 2X buffer D (300 mM NaCl, 20% NP40, 10% Sodium Deoxycholate, 2 mM EDTA, 2 mM EGTA, 100 mM Tris HCl, pH 7.5) was added and the sample was rotated for 1 h. Next, the sample was centrifuged at 20,800 × *g* for 20 min and the supernatant was transferred into a new falcon tube. The sample was then centrifuged for the second time using the same conditions, with the supernatant being collected. 100 µl was saved as input and protein concentration was quantified. 15 µl of GFP-trap beads (ChromoTek, #gmta-20) per mouse brain was added and the sample was incubated overnight. The next day, beads were washed three times (10 minutes each, 500 µl of 1X buffer D). After the final wash, the protein was eluted at 70 °C with 30 µl of 1X LDS-PAGE sample buffer (Thermo Fisher, #NP0007). 3 µl of the IP sample was used for SDS-PAGE and Silver staining (Invitrogen, #LC6070). The rest of the IP samples were run on a short SDS gel followed by in-gel digestion[82], the peptides were then labeled by TMT11 respectively, mixed equally, followed by desalting for the subsequent fractionation. Only 8 of the 11 channels' data was discussed here. All TMT channel information can be found in Supplementary Data 1a.

### Extensive two-dimensional liquid chromatography-tandem mass spectrometry
For each of the 4 TMT sets (iNeuron (16 samples, genotypes and clone numbers as described above), *Atg7* cKO mouse brain (3 *Atg7*[f/f]-SynCre and 4 *Atg7*[f/f] mice brains), *Atg14* cKO mouse brain (3 *Atg14*[f/f]-SynCre and 4 *Atg14*[f/f] mice brains) and GFP-LC3 affinity purification samples (described above). All the human iNeurons were generated from the three independent biological replicates (*n* = 3). All the mice were 6-8 weeks old, both male and female mice were used. Western blots using the remaining cell lysates after the Mass-spectrometry was used to confirm the genotype of the samples of all human iNeurons and mouse samples. The TMT labeled samples were fractionated by offline basic pH reverse phase LC respectively, and each of these fractions was analyzed by the acidic pH reverse phase LC-MS/MS[40]. The offline basic

pH LC was performed with an XBridge C18 column (3.5 µm particle size, 4.6 mm × 25 cm, Waters), buffer A (10 mM ammonium formate, pH 8.0), buffer B (95% acetonitrile, 10 mM ammonium formate, pH 8.0), using a 2–3 h gradient of 15–35% buffer B. In the acidic pH LC-MS/MS analysis, fractions were analyzed sequentially on a column (75 µm × 15–30 cm, 1.9 µm C18 resin from Dr. Maisch GmbH, 65 °C to reduce backpressure) coupled with a Fusion or Q Exactive HF Orbitrap mass spectrometer (Thermo Fisher Scientific). Peptides were analyzed with a 1–3 h gradient (buffer A: 0.2% formic acid, 5% DMSO; buffer B: buffer A plus 65% acetonitrile). For mass spectrometer settings, positive ion mode and data-dependent acquisition were applied with one full MS scan followed by a 20 MS/MS scans. MS1 scans were collected at a resolution of 60,000, $1 \times 10^6$ AGC and 50 ms maximal ion time; higher energy collision-induced dissociation (HCD) was set to 32–38% normalized collision energy; -1.0 m/z isolation window with 0.3 m/z offset was applied; MS2 spectra were acquired at a resolution of 60,000, fixed first mass of 120 m/z, 410–1600 m/z, $1 \times 10^5$ AGC, 100–150 ms maximal ion time, and -15 s of dynamic exclusion. For the phosphoproteomics analysis, 1.5 m/z isolation window was used to improve sensitivity.

### Protein and phosphopeptide identification and quantification by the JUMP software suite
The bioinformatics processing of protein and phosphopeptide identification and quantification were carried out with the JUMP software suites (Version 1.13.0)[83]. In brief, MS/MS raw data were searched against a target-decoy database to estimate false discovery rate (FDR)[84]. We combined the downloaded Swiss-Prot, TrEMBL, and UCSC databases and removed redundancy (human: 83,955 entries; mouse, 59,423 entries; date of downloading: April 20, 2020) to create the database. Main search parameters were set at precursor and product ion mass tolerance ( ± 15 ppm), full trypticity, maximal modification sites (*n* = 3), maximal missed cleavage (*n* = 2), static mass shift including carbamidomethyl modification (+57.02146 on Cys), TMT tags (+229.16293 for TMT10/11 or +304.20714 for TMT16 on Lys and N-termini), and dynamic mass shift for oxidation (+15.99491 on Met) and Ser/Thr/Tyr phosphorylation (+79.96633). Peptide-spectrum matches (PSM) were filtered by mass accuracy, clustered by precursor ion charge, and the cutoffs of JUMP-based matching scores (J-score and ΔJn) to reduce FDR below 1% for proteins during the whole proteome analysis or 1% for phosphopeptides during the phosphoproteome analysis. Protein and phosphopeptide quantifications were performed based on the reporter ions from MS2 using our previously optimized method[40].

The external solution used during recordings consisted of (in mM): 140 NaCl, 5 KCl, 10 HEPES, 2 CaCl$_2$, 2 MgCl$_2$, and 10 Glucose, with a pH of 7.4. All recordings of cell culture experiments were performed at room temperature.

To ensure data quality, any cells exhibiting a change in series resistance (Rs) exceeding 20% during the recording were excluded from the analysis. Additionally, recordings with an access resistance greater than 25 MΩ were also excluded. The electrophysiological data were collected using pClamp 10.5 software (Molecular Devices) and the MultiClamp 700B amplifier (Molecular Devices). The detection of miniature excitatory postsynaptic currents (mEPSCs) was performed using the template search algorithm in Clampfit 10.5 software (Molecular Devices).

### Quantification and statistical analysis
Unless otherwise indicated, all data presented are the average of at least three biological replicates from each of at least three independent experiments. The statistical significance of differences between two groups was determined using the unpaired two-tailed Student's *t* test or Mann–Whitney U test based on the normality test. For multiple-means comparisons, statistical significance was

determined by one-way analysis of variance followed by Dunnett's multiple comparisons test using GraphPad Prism 10 (GraphPad Software, CA, USA). All values are reported as mean ± SEM. See figure legends for details on specific statistical tests run for each experiment. Error bars were calculated using the standard deviation of all the replicates. Graphs and plots were generated using Graphpad Prism 10.0 or RStudio (2022.07.1). Figures were generated using Adobe Illustrator 2022.

## Reporting summary

Further information on research design is available in the Nature Portfolio Reporting Summary linked to this article.

## Data availability

The mass spectrometry proteomics data are available via ProteomeXchange with the identifier PXD048730. The authors declare that all data supporting the findings of this study are available within the paper and its Supplementary Information files. Further information and requests for resources and reagents should be directed to and will be fulfilled by the lead contact, Zhenyu Yue (zhenyu.yue@mssm.edu), Nan Yang (nan.yang1@mssm.edu) and Junmin Peng (junmin.peng@stjude.org). Source data are provided with this paper.

## Code availability

The source codes used for proteomics data analysis are available at https://github.com/JUMPSuite/JUMP. The code is citable from https://doi.org/10.5281/zenodo.10778288.

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

## Acknowledgements

We are grateful to Ruiqi Hu for technical supports, Dr. Martin Kampmann for the kind donation of i3N cell line. We are also grateful to Dr. Insup Choi, Edward Wickstead, George Heaton and Marianna Liang in Yue's laboratory for critical reading and discussion of the manuscript. The schematics were created using BioRender. This work was supported by the NIH grants R01NS060123 (Z.Y.), R01NS117590 and R21AG067570 (Z.Y. and N.Y.). X.Z. was supported by China Scholarship Council and Central South University, Xiangya Hospital.

## Author contributions

Zhenyu Yue conceived the study. Xiaoting Zhou generated all autophagy mutant lines of iNeurons, autophagy deficient (Atg7 and Atg14) mouse models, and performed autophagy assays in iNeurons and mouse brains. You-Kyung Lee performed the sample preparation and immunoflueoscence staining with *Atg7* cKO mice brains. Xianting Li performed the proteomics sample preparation, immunoblot assay and GFP affinity purification using mouse brains. Henry Kim assisted immunoflueoscence staining of iNeurons, making schematics and data quantification. Carlos Sanchez-Priego generated the H1-dcas9-krab cell line. Xian Han, Haiyan Tan, Suiping Zhou performed the proteomics and phospho-proteomics analysis by MS. Yingxue Fu assisted PPI protein network analysis. Kerry Purtell assisted the *Atg14* cKO mice breeding and data interpretation. Qian Wang performed the RNA sequencing data analysis, Gay Holstein performed the electron microscopy and helped with the EM analysis of the Atg7 cKO mouse brains. Beisha Tang guided the experimental design and discussion of the results. Junming Peng guided the proteomics data analysis and data interpretation. Zhenyu Yue, Xiaoting Zhou, You-Kyung Lee, Nan Yang, Junmin Peng drafted the manuscript.

## Competing interests

The authors declare no competing interests.

## Additional information

[1]Department of Neurology, The Friedman Brain Institute, Icahn School of Medicine at Mount Sinai, New York, NY 10029, USA. [2]Nash Family Department of Neuroscience, The Friedman Brain Institute, Icahn School of Medicine at Mount Sinai, New York, NY 10029, USA. [3]Department of Geriatrics, Xiangya Hospital, Central South University, Changsha 410008 Hunan, China. [4]Institute for Regenerative Medicine, Alper Center for Neural Development and Regeneration, Icahn School of Medicine at Mount Sinai, New York, NY 10029, USA. [5]Department of Structural Biology, St. Jude Children's Research Hospital, Memphis, TN 38105, USA. [6]Department of Developmental Neurobiology, St. Jude Children's Research Hospital, Memphis, TN 38105, USA. [7]Integrated Biomedical Sciences Program, University of Tennessee Health Science Center, Memphis, TN 38163, USA. [8]Center for Proteomics and Metabolomics, St. Jude Children's Research Hospital, Memphis, TN 38105, USA. [9]Department of Neurology, Xiangya Hospital, Central South University, Changsha 410008 Hunan, China. [10]National Clinical Research Center for Geriatric Disorders, Xiangya Hospital, Central South University, Changsha 410008 Hunan, China. [11]Center of Parkinson's Disease Neurobiology, The Friedman Brain Institute, Icahn School of Medicine at Mount Sinai, 1470 Madison Avenue, New York, NY 10029, USA. ✉e-mail: junmin.peng@stjude.org; nan.yang1@mssm.edu; zhenyu.yue@mssm.edu

