## [Peer Review File · Nature Communications]

Integrated proteomics reveals autophagy landscape and an autophagy receptor controlling PKA-R1 complex homeostasis in neuronEditorial Note: This manuscript has been previously reviewed at another journal that is not operating a transparent peer review scheme. This document only contains reviewer comments and rebuttal letters for versions considered at *Nature Communications*.

REVIEWER COMMENTS

Reviewer #1 (Remarks to the Author):

The authors have addressed my comments satisfactorily.

Reviewer #3 (Remarks to the Author):

The authors have addressed all my concerns and I have no further comments.

Reviewer #4 (Remarks to the Author):

Zhou et al present a revised version of their Ms on the proteomic analyses of autophagy-related substrates and other alteration in human iPS-derived neurons and brains from conditional KO mice. I remain enthusiastic about the comprehensiveness of the proteomic datasets provided in this paper which largely confirm recent observations from other labs on the ER as a major axonal substrate of autophagy in CNS neurons and mouse models.

Two key points raised in the previous round of review, however, in my opinion have not been appropriately addressed and/ or remain inconsistent:

1. The claimed effect of autophagy on SVs turnover remains far from compelling. In their rebuttal the authors provide EM data that argue against a significant effect of ATG7 loss on SV numbers. The biochemical data provided as a figure for the referee only show a tendency towards elevated levels of newly synthesized SV2A and Synaptogyrin3 over a chase period of 24h. Hence, both approaches have failed to yield conclusive evidence for a significant role of autophagy in SV turnover.

I therefore suggest that the EM data be incorporated into the revised Ms. Moreover, the results and discussion sections of the Ms would need to be re-written to make clear that no significant effects of atg7 loss on SV numbers or turnover could be detected. This certainly does not rule out that autophagy may contribute to the regulation of SV or SV protein turnover under specific conditions but the current dataset in my view provides very limited support for such a model.

2. The dataset pertaining to AKAP11 KD shown in Fig. 7 – as said before- clashes with a model according to which autophagy modulates cFos signaling via AKAP11-PKA. AKAP11 and PKA both accumulate in atg7 KO brains, which also display reduced cFos signaling. It appears now that no clear effects of AKAP11 loss on cFos signaling are seen, while PKA accumulates in absence of AKAP11 and this correlates with reduced mEPSC frequency. Clearly, these datasets are incompatible with a simple model, in which AKAP11 links PKA to autophagy and thereby regulates its activity. Given this major problem and the fact that the Ms focusses on the proteomic analysis of autophagy substrates in neurons, Fig. 7 should be removed from the paper and the discussion be edited accordingly.

Zhou et al present a revised version of their Ms on the proteomic analyses of autophagy-related substrates and other alteration in human iPSC-derived neurons and brains from conditional KO mice. I remain enthusiastic about the comprehensiveness of the proteomic datasets provided in this paper which largely confirm recent observations from other labs on the ER as a major axonal substrate of autophagy in CNS neurons and mouse models.

We are grateful to the reviewer's support to our manuscript.

Two key points raised in the previous round of review, however, in my opinion have not been appropriately addressed and/ or remain inconsistent:

1. The claimed effect of autophagy on SVs turnover remains far from compelling. In their rebuttal the authors provide EM data that argue against a significant effect of ATG7 loss on SV numbers. The biochemical data provided as a figure for the referee only show a tendency towards elevated levels of newly synthesized SV2A and Synaptogyrin3 over a chase period of 24h. Hence, both approaches have failed to yield conclusive evidence for a significant role of autophagy in SV turnover.

I therefore suggest that the EM data be incorporated into the revised Ms. Moreover, the results and discussion sections of the Ms would need to be re-written to make clear that no significant effects of atg7 loss on SV numbers or turnover could be detected. This certainly does not rule out that autophagy may contribute to the regulation of SV or SV protein turnover under specific conditions but the current dataset in my view provides very limited support for such a model.

We are sorry that we have not clarified this point in our previous revision. In fact, our view is completely in line with the reviewers on the notion that autophagy may not turn over SVs. Our conclusion is that autophagy plays a role in maintaining the protein homeostasis of a set of SV proteins likely through autophagy degradation. As suggested by the reviewer, we added the EM result (Supplementary Figure 3L) in the manuscript, which argues that autophagy does not directly turn over SVs, and edited the related part in the Discussion (See discussion, the third paragraph yellow highlighted).

2. The dataset pertaining to AKAP11 KD shown in Fig. 7 – as said before- clashes with a model according to which autophagy modulates cFos signaling via AKAP11-PKA. AKAP11 and PKA both accumulate in atg7 KO brains, which also display reduced cFos signaling. It appears now that no clear effects of AKAP11 loss on cFos signaling are seen, while PKA accumulates in absence of AKAP11 and this correlates with reduced mEPSC frequency. Clearly, these datasets are incompatible with a simple model, in which AKAP11 links PKA to autophagy and thereby regulates its activity. Given this major problem and the fact that the Ms focusses on the proteomic analysis of autophagy substrates in neurons, Fig. 7 should be removed from the paper and the discussion be edited accordingly.

We agree with the reviewer on the insightful comment. Thus, we removed the Figure 7 E and F, which showed that loss of AKAP11 results in the reduction of mEPSC frequency and impairment of neurotransmission. But we would like to keep the data demonstrating that lack of AKAP11 causes accumulation of PKA RI and Ca subunits (Figure 7 A-D), consistent with its function in mediating the clearance of PKA-RI complex through autophagy in neurons. We agree that the removal of Figure 7E/F keeps the manuscript more focused and easier to follow. Accordingly, we revised our manuscript as shown in the Title, Abstract, Introduction, Results and Discussion with yellow highlight.

REVIEWERS' COMMENTS

Reviewer #4 (Remarks to the Author):

My final comments have been appropriately addressed and I support publication of this important Ms in Nature Communications.